# The impacts of antipsychotic medications on eating-related outcomes: A mixed methods systematic review

Rasha Alkholy[1,2,3]*, Karina Lovell[1,2,3], Penny Bee[1,2,3], Rebecca Pedley[1,2,3], Helen Louise Brooks[1,2], Richard J. Drake[2,4,5], Prathiba Chitsabesan[3,6,7], Anam Bhutta[1,2,3], Abigail Brown[1], Rebecca L. Jenkins[1,2,3], Andrew Grundy[1,2,3]

1 Division of Nursing, Midwifery and Social Work, School of Health Sciences, Faculty of Biology, Medicine and Health, The University of Manchester, Manchester, United Kingdom, 2 Manchester Academic Health Science Centre, The University of Manchester, Manchester, United Kingdom, 3 National Institute for Health and Care Research Applied Research Collaboration (NIHR ARC) Greater Manchester Mental Health Theme, Manchester, United Kingdom, 4 Division of Psychology and Mental Health, Faculty of Biology, Medicine and Health, The University of Manchester, Manchester, United Kingdom, 5 Greater Manchester Mental Health NHS Foundation Trust, Manchester, United Kingdom, 6 Pennine Care NHS Foundation Trust, Ashton-under-Lyne, Greater Manchester, United Kingdom, 7 Faculty of Psychology, University College London, London, United Kingdom

* Rasha.Alkholy@manchester.ac.uk

## Abstract

### Background

Almost all antipsychotics are associated with weight gain. Given the gravity of this side-effect and its consequences, it is imperative to understand the mechanisms involved. One mechanism that could contribute to this side effect is the impact of antipsychotics on eating-related outcomes.

### Objective

We aimed to synthesise the available quantitative research on the effects of first- and second-generation antipsychotics on eating-related outcomes, and qualitative research exploring people's experiences with these medications in relation to appetite and eating behaviours (PROSPERO protocol CRD42022340211).

### Methods

We searched Medline, PsycInfo, and Web of Science from inception to 9 May 2024. Quantitative data were synthesised without meta-analysis using vote counting based on direction of effect. Qualitative data were synthesised using thematic synthesis.

### Results

Searches identified 8,746 citations yielding 61 separate studies; 55 quantitative and 6 qualitative, published 1982–2024. Using GRADE, our assessment of the quantitative review findings ranged from low to very low-level certainty. Given the lack of direct

**Data availability statement:** All relevant data are within the paper and its Supporting Information files.

**Funding:** This paper presents research funded by the National Institute for Health and Care Research (NIHR) Applied Research Collaboration Greater Manchester (ARC-GM); Grant number NIHR200174 (https://arc-gm. nihr.ac.uk/mental-health[arc-gm.nihr.ac.uk]). KL is a co-applicant on the ARC award and the Mental Health Theme Lead. PB is the Deputy Theme Lead. The views expressed are those of the authors and not necessarily those of the NIHR or the Department of Health and Social Care. The funders had no role in study design, data collection and analysis, decision to publish, or preparation of the manuscript.

**Competing interests:** The authors have declared that no competing interests exist.

evidence from high-quality placebo-controlled trials, it is pertinent to interpret the quantitative findings with caution. Using GRADE-CERQual, our assessment of the qualitative review findings ranged from low to very low-level certainty; these findings suggest that the relationship between antipsychotics and food intake is influenced by an interplay of individual, interpersonal and external factors, the most significant of which is food environment.

## Limitations

The internal validity of this review was affected by the serious limitations of the included quantitative studies and the paucity of qualitative evidence.

## Strengths

We used GRADE and GRADE-CERQual frameworks to enhance the transparency of our judgement of the certainty of the evidence. Lived experience perspectives were incorporated in different stages of the review to enhance its relevance and practical implications.

## Conclusions

There is insufficient evidence from well-conducted studies to determine the effect of antipsychotics on eating-related outcomes.

## 1. Introduction

Antipsychotic medications are the foundation of treatment for non-affective psychoses, such as schizophrenia, schizoaffective disorder and delusional disorder [1]. They are also widely used for bipolar disorder [2], delirium [3], psychotic and treatment-resistant depression [4]. Nevertheless, these medications are associated with a wide array of side effects, some of which are serious and have been implicated in reduced concordance [5], the consequences of which include relapse, rehospitalisation [6], longer time to remission, and attempted suicide [7]. One of these serious side-effects is weight gain. Numerous meta-analyses [1,8–13] and network meta-analyses [14,15] show that nearly all antipsychotics are associated with weight gain, with clozapine and olanzapine associated with the most severe weight gain and aripiprazole the least. The extent of induced weight gain varies according to the receptor binding affinity of these medications. Evidence suggests that antipsychotics with higher histaminergic (H1), serotonergic (5-HT2C), cholinergic (M1 and M3) receptor affinity are associated with a higher risk of weight gain and obesity compared to other antipsychotics [16].

Obesity is a major precursor for and a core criterion of the metabolic syndrome- a complex syndrome defined by a cluster of risk factors that are associated with cardiovascular disease [17] and excess all-cause mortality [18]. Other aspects of that syndrome include dyslipidaemia, impaired glucose tolerance and hypertension [19], all of which are also direct metabolic adverse-effects of antipsychotic medications [20,21]. Together, antipsychotic-induced weight gain, obesity and metabolic side effects partly explain the shorter life expectancy and higher mortality rates among people with serious mental illnesses compared to the general population [22,23].

Given the gravity of antipsychotic-induced weight gain and its consequences, it is imperative to understand the mechanisms involved. Several factors are believed to contribute to this side-effect including pretreatment and premorbid genetic vulnerabilities, socioeconomic

disadvantages, sedentary lifestyle, and unhealthy eating behaviours [24]. Focusing on potentially modifiable factors, such as unhealthy eating behaviours, provides an avenue to tackling, or at least restricting, antipsychotic-induced weight gain and its sequelae.

Unhealthy eating behaviours associated with antipsychotics have been the focus of two recently published reviews: a scoping review [25] and a systematic review [26]. The scoping review [25] did not assess the risk of bias of the results of included studies. Thus, the resulting narrative synthesis did not take into account the limitations of the included studies and their impact on the certainty of the review findings and their interpretation. Additionally, although the scoping review focused on eating behaviours in people with psychosis spectrum disorders, it included a number of studies in which the majority (> 75%) of participants did not receive antipsychotics. The systematic review and meta-analysis [26] investigated the effect of second-generation antipsychotics, but included a heterogeneous population of participants in terms of age groups and clinical diagnoses. The age range of the participants, reported by 83 of the 92 included studies, was 2 to 85 years. These studies included participants with a vast array of mental disorders, such as bipolar disorder, affective disorders, alcohol dependence disorder, delirium and dementia; all of which have been shown themselves to affect appetite. The authors used meta-analyses to combine effect estimates of distorted eating cognitions derived from three cross-sectional studies [27–29]. However, these studies were at high risk of selection bias [27–29], used non-comparable comparison groups [27,28] and did not account for confounding [27–29]. Together these factors may complicate the interpretation of the results of both reviews. Moreover, these reviews did not examine qualitative evidence on the effect of antipsychotics on the nature of hunger and eating behaviour, which is crucial if we are to gain insight into people's experiences and beliefs and develop a deeper understanding of the different factors that are perceived to influence these experiences.

The aim of our review was to synthesise quantitative and qualitative research on eating-related outcomes of taking first- and second-generation antipsychotics using a more diagnostically homogeneous sample than previous reviews and making fewer assumptions during quantitative synthesis. To facilitate the presentation and interpretation of the findings of this review, eating-related outcomes were categorised into four core dimensions: i) appetite sensations (appetite, hunger, fullness and/or satiety, craving); ii) food intake and dietary composition; iii) eating behaviours and cognitions (e.g., dietary restraint, disinhibition, hunger); iv) eating disorders.

Review questions:

- What eating-related outcomes have been investigated in adult populations prescribed antipsychotics and how have these been measured?

- How do antipsychotics affect eating-related outcomes?

- What are the experiences of people using antipsychotics in terms of their impact on eating behaviours?

## 2. Methods

The protocol was pre-registered in PROSPERO (registration number CRD42022340211). Reporting of this review followed the Preferred Reporting Items for Systematic Reviews and Meta-Analyses (PRISMA) [30] (S1 File) and the Enhancing transparency in reporting the synthesis of qualitative research (ENTREQ) [31] guidelines.

For the purposes of this review, the term 'patients' refers to participants with clinical diagnoses of mental health problems and the term 'healthy volunteers' refers to participants without clinical diagnoses of mental health problems.

## 2.1. Methods for identifying primary studies to be included in the systematic review

**2.1.1. Eligibility criteria.** The full eligibility criteria are presented in Fig 1.

**2.1.2. Information sources and search strategy.** We conducted a systematic search of three electronic databases (Medline (OVID interface), PsycInfo (OVID interface), and the Web of Science Core Collection) from database inception to 9 November 2021, updated on 15 March 2023 and 9 May 2024. To supplement the electronic databases search, reference lists of included study records and relevant systematic reviews [25,26] were checked to identify further eligible records. The search terms addressed two key conceptual areas 'eating indicator or behaviour' and 'antipsychotics'. Details of the search strategy are provided in the S2 File.

**2.1.3. Selection process.** Two authors independently screened titles and abstracts, and subsequently full texts against the pre-defined eligibility criteria; disagreements were resolved by a third author (RA, KL, PB, RP, HLB, AB, ABr, RLJ, AG). A list of full-text records meeting the eligibility criteria was compiled and subjected to data extraction and quality appraisal.

| | Inclusion criteria | Exclusion criteria |
|---|---|---|
| **Participants** | People aged 16 years or over who had been prescribed first- or second-generation antipsychotic medications (either patients with clinical diagnoses that required the prescription of antipsychotics or healthy volunteers who received medications during the course of a study). Studies including people with mild-moderate learning disabilities who had been prescribed antipsychotics. Studies exploring perspectives of carers/family members supporting people prescribed first- or second-generation antipsychotic medications. | Focused on children and adolescents aged under 16. < 75% of the exposed group (i.e., cases) were prescribed antipsychotics. >25% of the sample were diagnosed with an eating disorder, autism, dementia, affective disorders, delirium. > 25% of the sample were diagnosed with a severe learning disability. Involved animals. |
| **Study outcome/ Phenomenon of interest** | Eating-related outcomes or experiences in people prescribed antipsychotic medications. Eating-related outcomes may be objectively or subjectively measured. They could be self-reported and/or proxy-reported by others (carers/family members supporting people prescribed antipsychotic medications). | Focused on eating-related outcomes in relation to medications other than antipsychotics. |
| **Context** | Studies conducted in any country. | |
| **Study design** | Randomised studies, non-randomised studies, quantitative descriptive and qualitative studies | |
| **Report characteristics** | English Language records. Published records. | Non-primary research |

**Fig 1. Eligibility criteria applied in the systematic review.**

**2.1.4. Data collection process.** Data were extracted by one reviewer using a pilot-tested data extraction form (S3 File) and verified by another; disagreements were resolved by discussion (RA, AG). Corresponding authors were contacted when insufficient data were reported in their studies. When multiple records used data collected from the same study sample, additional relevant data were extracted from all records.

**2.1.5 Study risk of bias assessment.** Risk of bias assessments were undertaken by one reviewer and checked by another using the Mixed Methods Appraisal Tool [32]; any discrepancies were resolved by consensus (RA, AG). The domains of the Mixed Methods Appraisal Tool [32] were not weighted and papers were not excluded based on a priori quality thresholds. The Grading of Recommendations, Assessment, Development, and Evaluations (GRADE) framework [33] provided a transparent and structured approach for assessing the certainty in quantitative evidence for six key outcomes: changes in appetite sensations; daily energy intake; dietary restraint, restriction and hunger; and eating disorders. The Grading of Recommendations, Assessment, Development, and Evaluations framework for Confidence in Evidence from Reviews of Qualitative research (GRADE-CERQual) [34] framework was used to assess confidence in key qualitative review findings.

## 2.2. Synthesis method

A parallel-results convergent design [35] was used. The extracted quantitative and qualitative data were synthesised and are presented separately in the 'Results' section. The resulting evidence was then integrated and interpreted collectively; located in the 'Discussion' section.

**2.2.1. Quantitative synthesis method.** Due to lack of direct evidence from high-quality placebo-controlled trials, we had to include less favourable evidence from within-group comparisons and studies of healthy volunteers. Thus, the included studies were grouped by type of synthesis (i.e., within-group or between-group differences), core eating-related outcome domain under investigation, type of participants (i.e., included antipsychotic-treated patients as one of the comparison groups or included healthy volunteers only) and study design (i.e., randomised controlled trials (RCTs) or non-randomised studies).

The synthesis focusing on within-group differences was restricted to comparative studies which investigated the effect of antipsychotics by assessing change in eating-related outcomes from baseline (i.e., pre- and post-treatment change score) in antipsychotic-treated participants. The synthesis focusing on between-group differences was restricted to comparative studies which investigated the effect of antipsychotics on eating-related outcomes by comparing antipsychotic-treated participants (i.e., exposed group) with untreated participants (i.e., unexposed group). Data extracted from studies of healthy volunteers were presented separately due to differences between this population and the target population of patients receiving antipsychotics. In synthesising the evidence, data extracted from randomised trials were analysed separately from data extracted from non-randomised studies as the latter study designs were expected to be systematically different and at higher risk of bias than randomised trials [36].

Aggregating effect estimates in each of the within-group and between-group syntheses was not possible due to several factors. Limited evidence was extracted per eating-related outcome for comparison due to the diversity of outcomes examined across the studies, differences in the operationalistion of the same outcome across studies and the incomplete reporting of outcomes in individual studies. Moreover, the included studies differed in terms of study design and risk of bias. Therefore, the evidence was synthesised using vote counting based on direction of effect [37], a method that entails the transformation of each effect estimate into a standardised binary metric, a positive or negative effect, based on the observed direction of effect alone. The number of positive effects is then compared to the number of negative

effects for each outcome. This disregard of 'statistical significance' allows the synthesis of data derived from underpowered studies, an advantage over the conventional forms of vote counting [37]. Nonetheless, this method neither provides information on the magnitude of effects nor accounts for differences in the quality or relative sizes of the studies [38]. Visual presentations of the data were created using effect direction plots [39]. The methods used to prepare the data for synthesis, including data conversion and handling of missing summary statistics, are provided in the S4 File.

Data on the prevalence of antipsychotic-induced changes in eating-related outcomes and antipsychotic-related factors associated with these changes (e.g., head-to-head comparisons between antipsychotics, effect of different doses) were summarised in narrative format (S5 File).

**2.2.2. Qualitative synthesis method.** Extracted qualitative data were synthesised using thematic synthesis [40], an inductive approach based on the principles of thematic analysis used in primary research. It shares characteristics with both the later adaptations of meta-ethnography and grounded theory [41]. Thematic synthesis is a three-staged process involving the iterative movement between line-by-line coding of text, identification of descriptive themes and generation of higher-level analytical themes [40].

The qualitative data extracted from all studies formed the qualitative dataset for this review. Each line of text within that dataset was coded based on its meaning and content. Groups of related codes were then categorised into broader descriptive themes. This was an iterative process that involved moving back to the original studies to ensure that the themes were coherent and grounded in the experiences of the study participants [40]. The final stage involved identifying conceptual links between descriptive themes and generating higher-level analytical themes, when applicable. This allowed the interpretation of the descriptive themes in light of the review questions, thus 'going beyond' the content of the included studies [40]. Each stage was conducted by one reviewer and verified by a second reviewer. Participant quotations and author interpretations were given equal weights in this review. Qualitative data were managed and analysed using NVivo12 [42].

**2.2.3. Modification to the protocol.** After protocol completion (PROSPERO registration number CRD42022340211) and references identification, but before synthesis, an additional criterion was introduced: exclusion of studies in which > 25% of the sample was diagnosed with affective disorders or delirium because of the significant effects these disorders have on eating-related outcomes and weight [43,44].

**2.2.4. Lived Experience and Public Involvement.** AG, a Lived Experience Researcher, was involved in all stages of the review from inception. Two Public Involvement Co-researchers (AB and RJ) were involved in title and abstract and full-text screening. AB contributed to manuscript review and editing. We involved our Lived Experience Advisory Panel (LEAP; n = 6; 4 service users, 2 carers) at the data synthesis stage to provide their interpretation of the findings. A Systematic Review Workshop was held with the LEAP in which the aims of and stages involved in conducting a systematic review were explained and the preliminary review findings were presented. The members were asked for their interpretation of the findings and whether these findings resonated with their own experiences with antipsychotic medications. They were then asked whether they would be interested in co-writing a systematic review commentary. Two members expressed interest and had a separate meeting with the reviewers at which the process of co-writing a commentary was explained and the final review findings were presented. The two members collaborated on writing the commentary and had two further meetings with the reviewers for feedback. Thus, the overarching aim of incorporating lived experience perspectives in this review was to ensure the review questions were relevant to the target population, to enhance the interpretation of the review findings, and to improve the practical impact of the conclusions drawn [45].

## 3. Results

### 3.1. Flow of studies through the review

A total of 8,746 citations were retrieved, imported and deduplicated with Endnote and Covidence [46]. After removal of duplicates, the number was reduced to 2,078. Of these, 1,911 were excluded at the title/abstract stage and 167 were retrieved for full-text examination. A list of all studies retrieved for full-text examination, including those that were excluded from the analyses and reasons for exclusion is provided in the S6 File. Fig 2 summarises the flow of studies and exclusion reasons for citations at the full-text screening phase.

Sixty-one citations met the inclusion criteria. These represented 61 separate studies reported across 65 papers; 55 quantitative and 6 qualitative studies. One study [47], described as a 'qualitative assessment', presented data in the form of prevalence rates of treatment-emergent adverse events (TEAEs) associated with second-generation antipsychotics. Data from this study were extracted and reported alongside other quantitative descriptive studies.

## 4. Results derived from quantitative studies

### 4.1. Overview of quantitative studies

Study and participant characteristics of the 55 included quantitative studies are provided in the S5 File. The studies varied in terms of study design, study population and the

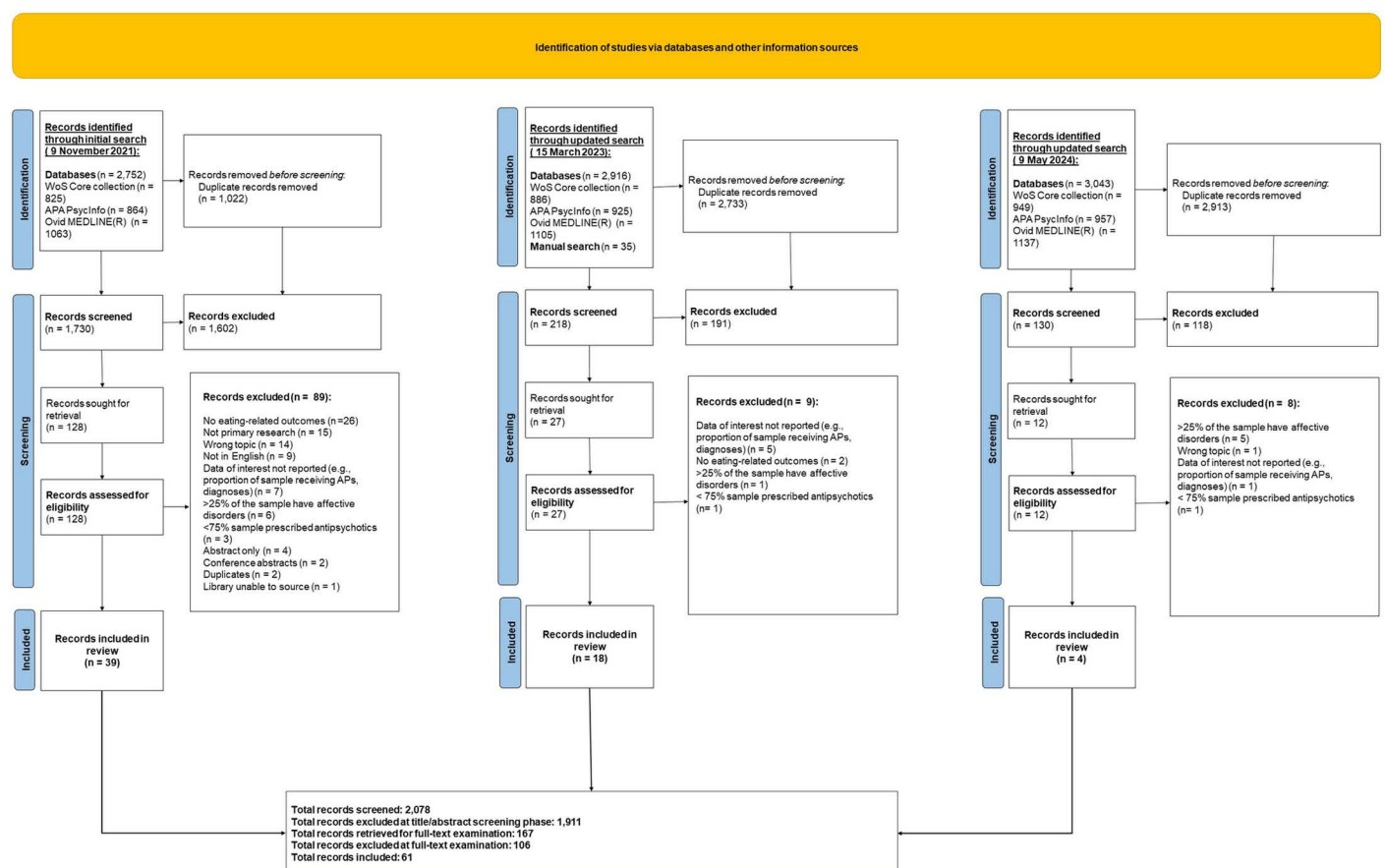

**Fig 2. Flow of studies through the review.** APA = American psychological association; n = number; WoS = web of science.

operationalisation of eating-related outcomes. Nineteen of these studies reported data from Europe [28,29,48–64], 15 from North America [27,47,65–77], nine from Asia [78–86], four from Australia [87–90], one from South America [91] and one from the Middle East [92]. Six studies reported data collected from several countries [93–98]. Eighteen were randomised controlled studies (reported across 20 papers) [48–50,65–72,78,85,86,93–96,99,100], 26 were non-randomised studies (reported across 28 papers) [27–29,51–59,63,64,73–77,79–81,87,91,97,98,100,101] and 11 were quantitative descriptive studies [47,60–62,82–84,88–90]. Publication dates ranged from 1982 to 2024.

The sample sizes ranged from 1 to 3,362 participants and included a total of 13,502 participants. Of the 13,502 participants, 3,402 had a diagnosis of schizophrenia, 282 schizoaffective disorder, 28 schizophreniform disorder, 4 schizotypal disorder, 123 bipolar disorder, 35 other related psychotic disorders (including delusional disorder, unspecified non-organic psychosis, depression, personality disorders), and 2,661 participants were collectively described as having schizophrenia or schizoaffective disorder. A total of 6,967 were healthy participants. The mean age of the patients included ranged from 17 to 51 years, while that of the healthy participants ranged from 25.4 to 44 years. Patients were recruited from inpatient and outpatient settings. Data on ethnicity were not consistently reported.

Olanzapine was the most commonly administered antipsychotic (n = 2,412) followed by clozapine (n = 790) and haloperidol (n = 770). Other antipsychotics included risperidone (n = 248), quetiapine (n = 143), aripiprazole (n = 126), amisulpride (n = 63), zuclopenthixol (n = 39), flupentixol (n = 19), paliperidone (n = 16), ziprasidone (n = 14), iloperidone (n = 7), pipotiazine (n = 4), lurasidone (n = 2), fluphenazine (n = 4), and asenapine (n = 1). Other studies categorised participants into those receiving first-generation antipsychotics (n = 267), second-generation antipsychotics (n = 727) or both (n = 42). One study [51] categorised participants into those receiving antipsychotics with a lower risk (n = 11) or higher risk of weight gain (n = 10), while three studies [58–60] did not provide details of the prescribed antipsychotics (n = 389). Duration of antipsychotic treatment was reported in 17 studies only and ranged from 19.90 weeks [94] to 30 years [84]. Five randomised studies [65,67,68,70,72,99], reported in six records, and 1 pre-post study [74] included healthy volunteers only. There was a wide variation in terms of the operationalisation and measurement of eating-related outcomes across the studies. These outcomes were measured in natural or laboratory settings using different questionnaires/scales and techniques (presented in the S7 File).

## 4.2. Risk of bias assessment

Appraising the quality of the RCTs was undermined by deficiencies in their reporting. Most of these studies did not provide adequate information about the methods used for allocation sequence generation and/or allocation concealment (n = 16/18) [49,50,65–72,78,85,93–96,99], blinding n = (8/18) [50,65–68,70,94,96] and the measures used to affirm participant adherence to the assigned intervention (n = 10/18) [49,50,66,68,70–72,78,95,96,99]. Five RCTs were unblinded [71,78,85,93,95]. The studies had high rates of attrition (n = 6/18) [65,67,68,70,93,96], used non-validated measures to assess outcomes (n = 12/18) [50,65,66,69–72,78,85,93,94,96,99], and did not provide complete outcome data (n = 10/18) [49,65–70,93,95,96].

Most of the non-randomised studies used non-probability sampling (n = 12/26) [28,29,52,53,57,63,73–75,81,87,91,101] rendering them at high risk of selection bias. Twelve studies [27,51,54,58,59,64,76,77,79,80,97,98] did not report their sampling techniques. Only three studies [63,75,101] investigated differences between responders and non-responders. In most studies, data from the antipsychotic-treated group were compared with data from working

adults/healthy volunteers (n = 11/26) [27,28,54,57–59,63,73,75,81,91] or a nationally representative sample of the general population (n = 3/26) [53,56,77], rather than a sample representative of the target population (e.g., antipsychotic-naïve patients). Only six studies, reported in seven records, controlled for baseline body mass index/weight using multivariate analysis [27,29,57] or stratification [28,52,63,101]. Restriction was used by eight studies to control for antipsychotic treatment duration [53,55,79], category [55,74,79,80,87] and sex [27,53,59,80]. Nonetheless, none of the studies controlled for all potentially significant confounders.

Similarly, most of the quantitative descriptive studies showed high risk of bias with seven of 11 studies using convenience sampling [47,88–90] or inadequately reporting on sampling strategies [61,82,83]; 10 studies at high risk of non-response bias [89,90,92] or inadequately reporting on non-responders [47,60–62,82,83,88]; and seven studies [47,61,62,83,89,90,92] using unclear or non-validated outcome measures.

The risk of bias assessments of the 55 included quantitative studies is provided in the S8 File.

## 4.3. Results of the quantitative synthesis

In the following section, the availability, direction of effect (regardless of statistical significance) of antipsychotic medications (exposure) on eating-related outcomes (outcome) and overall GRADE quality of evidence are presented, grouped by type of synthesis (i.e., within-group or between-group differences), core eating-related outcome domain under investigation, type of participants and study design. Effect direction plots provide a visual presentation of the data.

GRADE Summary of Findings Tables for key outcomes for within-group and between-group syntheses are provided in the respective results sections. A detailed account of the quantitative synthesis and the GRADE Evidence Profile for each of the included outcomes is available in the S9 File and S10 File, respectively.

### 4.3.1. Results of within-group syntheses.

a) Appetite sensations

Fig 3 shows the effect direction plot for this section of the results.

*Appetite as a composite variable:* Four RCTs [66,71,78,94], including a total of six independent samples (4 of high baseline BMI, 2 of normal baseline BMI), assessed appetite change using different appetite measurement scales (i.e., included different combinations of hunger, satiety, desire to eat or prospective food consumption), over different time intervals ranging from two weeks [94] to five months [71]. Two of these studies [66,71] assessed appetite, craving and eating cognitions collectively. Only two studies had washout periods, ranging from three days [66] to three months [78], while in the study by Karagianis and colleagues [94] participants had already been receiving olanzapine for 4 to 52 weeks (mean = 19.9 weeks) at enrolment. Three of four samples [71,94], all with high baseline BMI, receiving olanzapine (2 samples) or risperidone (1 sample), reported a decline in appetite from baseline. Two samples [78], with normal baseline BMI, receiving ziprasidone or olanzapine, reported no effect.

A prospective, observational study including olanzapine-treated patients with normal baseline BMI [98] reported a decline in appetite scores over a four-week period. This study assessed appetite and eating cognitions collectively.

Using GRADE, there was very low-level RCT evidence that antipsychotics (olanzapine, risperidone) decreased appetite in patients with high baseline BMI; very low-level RCT evidence that antipsychotics (olanzapine, ziprasidone) had an inconsistent effect on patients with normal baseline BMI; and very low-level observational study evidence that antipsychotics (olanzapine) decreased appetite in patients with normal baseline BMI.

| BMI | Citation | Study design | End of study assessment | Sample size (n) | Antipsychotic | Measurement scale | Washout period (days) | Appetite (composite) | Craving — General food craving | Complex carbohydrates/proteins | Simple sugar/trans fat | Fast-food fats (FFF) | Hunger |
|---|---|---|---|---|---|---|---|---|---|---|---|---|---|
| **QUANTITATIVE RCT EVIDENCE** | | | | | | | | | | | | | |
| High | Case et al., 2010 Study 1 (Hardy et al., 2009) | 2-arm parallel, double-blind RCT | 2 weeks | 68 | Olanzapine | EBA a,b | 3-10 | ▲ | | | | | |
| | Case et al., 2010 Study 2 (Karagianis et al., 2009) | 2-arm parallel, double blind RCT | 2 weeks | 65 | Olanzapine | PARS a | No | ▼ | | | | | |
| | Case et al., 2010 Study 3 (Hoffman et al., 2009) | 3-arm parallel, open label RCT | 2 weeks | 50 | Olanzapine | FCI | NR | | ▼ | | | | |
| | (Smith et al., 2012) | 2-arm parallel, open-label RCT | 5 months | 23 | Olanzapine | VAS a EBA a,b | No | ▼ | | | | | |
| | (Smith et al., 2012) | 2-arm parallel, open-label RCT | 5 months | 23 | Risperidone | VAS a EBA a,b | No | ▼ | | | | | |
| Normal | (Park et al., 2013) | 2-arm parallel open-label RCT | 12 weeks | 10 | Ziprasidone | VAS a | 90 | ◄► | | | | | |
| | (Park et al., 2013) | 2-arm parallel open-label RCT | 12 weeks | 10 | Olanzapine | VAS a | 90 | ◄► | | | | | |
| **Overall quality of RCT evidence** | | | | | | | | ⊕○○○ | ⊕⊕○○ | | | | |
| **QUANTITATIVE NON-RANDOMISED STUDY EVIDENCE** | | | | | | | | | | | | | |
| High | (Garriga et al., 2019) | Cohort | 18 weeks | OWO group: 21 | Clozapine | FCI-SP | NR | | ▼ | ▲ | ▼ | ▼ | |
| Normal | Case et al., 2010 Study 4 (Treuer et al., 2009) | Secondary analysis of a prospective 6-month study | 4 weeks | 606 | Olanzapine | EAS a,b | NR | ▼ | | | | | |
| | (Garriga et al., 2019) | Cohort | 18 weeks | NW group: 13 | Clozapine | FCI-SP | NR | | ▲ | ▲ | ▲ | ▲ | |
| **Overall quality of quantitative non-randomised study evidence** | | | | | | | | ⊕○○○ | ⊕○○○ | | | | |
| **STUDIES OF HEALTHY VOLUNTEERS** | | | | | | | | | | | | | |
| Normal | (Ballon et al., 2018) | 3-arm double-blind, parallel RCT | 28 days | 7 | Olanzapine | VAS a | NA | | | | | | ▲ |
| | (Ballon et al., 2018) | 3-arm double-blind, parallel RCT | 28 days | 7 | Iloperidone | VAS a | NA | | | | | | ▲ |
| | (Roerig et al., 2005) | 3-arm parallel, double-blind RCT | 2 weeks | 16 | Olanzapine | VAS a | NA | | | | | | ▲ |
| | (Teff et al., 2013) | 3-arm parallel, double-blind RCT | 12 days | 10 | Olanzapine | VAS a | NA | | | | | | ▲ |
| | (Teff et al., 2013) | 3-arm parallel, double-blind RCT | 12 days | 10 | Aripiprazole | VAS a | NA | | | | | | ▲ |
| **Overall quality of evidence from studies of healthy volunteers** | | | | | | | | | | | | | ⊕○○○ |

**Fig 3. Within-group syntheses - Effect direction plot of antipsychotics on appetite sensations.** a = unvalidated measurement scale; b = collective scale (i.e., measured appetite and food craving or eating cognitions collectively). BMI = Body mass index; EAS = Eating Apppetite Scale/Eating Attitude Scale; EBA = Eating Behavior Assessment; FCI = Food Craving Inventory; FCI-SP = Food Craving Inventory-Spanish Version; n = number; NA = not applicable; NR = not reported; NW = normal weight; OWO = overweight/obese; PARS = Platypus Appetite Rating Scale; RCT = randomised controlled trial; VAS = Visual analogue scale. Effect direction: upward arrow ▲ = positive effect; downward arrow ▼ = negative effect; sideways arrow ◄► = no change/mixed effect. Sample size: large arrow ▲ ≥ 300; medium arrow ▲ ≥ 100; small arrow ▲ 50-99; very small arrow ▲ < 50. Overall quality of evidence: ⊕○○○ very-low quality; ⊕⊕○○ low quality.

*Hunger:* The impact of olanzapine, iloperidone and aripiprazole on hunger was assessed in a laboratory setting in three RCTs of healthy volunteers with normal baseline BMI who were followed up for 28, 14 and 12 days, respectively [65,70,72]. They all reported an increase in hunger scores from baseline. Using GRADE, there was very low-level RCT evidence that antipsychotics increased hunger among healthy volunteers with normal baseline BMI.

*Food cravings:* One RCT [95], including participants with high baseline BMI, reported a decline in general food craving scores (using the Food Craving Inventory) over a two-week olanzapine treatment.

A prospective, observational study [52], including two independent samples stratified by baseline BMI, investigated the effect of an 18-week clozapine treatment on food cravings. Normal weight patients experienced a consistent increase in general and specific food cravings from baseline, whereas overweight or obese patients experienced a decline in general and specific food carvings, except for complex carbohydrates/proteins [52].

Using GRADE, there was low-level RCT evidence (olanzapine) and very low-level observational study evidence (clozapine) that antipsychotics decreased general food cravings among

patients with high baseline BMI. Observational study evidence showed that antipsychotics increased cravings among patients with normal baseline BMI.

b) Food intake and dietary composition

Fig 4 shows the effect direction plot for this section of the results.

Only one observational, prospective study [80] examined the change in energy and macronutrient intake among olanzapine-treated patients with normal baseline BMI (after a mean washout period of 17.6 days). It reported an increase in energy and total carbohydrate intake and a decrease in total fat and protein intake over 4 weeks. Using GRADE, there was very low-level observational study evidence that olanzapine increased energy intake among patients with normal baseline BMI.

Three RCTs of healthy volunteers with normal baseline BMI [65,70,99], including a total of six independent samples, found an increase in energy intake from baseline in olanzapine-treated participants. However, a contradictory effect was observed in healthy volunteers receiving iloperidone [65], risperidone [70] and aripiprazole [99]. Two RCTs of healthy volunteers [65,67] found an increase in protein intake from baseline in olanzapine-treated participants, regardless of baseline BMI. Using GRADE, there was very low-level evidence that olanzapine increased energy intake among healthy volunteers with normal baseline BMI.

c) Eating cognitions and behaviours

Fig 5 shows the effect direction plot for this section of the results.

Two RCTs [86,95] examined the effect of a 14-day and 5-day olanzapine treatment on eating behaviours using the Three-Factor Eating Questionnaire (TFEQ) and the Three-Factor Eating Questionnaire-Revised 21-Item Version (TFEQ-R21), respectively. Both studies showed a consistent decline across all domains of the TFEQ and TFEQ-R21 from baseline,

| BMI | Citation | Study design | End of study assessment | Sample size (n) | Antipsychotic | Measurement scale | Washout period (days) | Energy | Total fat | Total carbohydrates | Total proteins |
|---|---|---|---|---|---|---|---|---|---|---|---|
| **QUANTITATIVE NON-RANDOMISED STUDY EVIDENCE** | | | | | | | | | | | |
| Normal | (Gothelf et al., 2002) | 4-week prospective study | 4 weeks | 10 | Olanzapine | Energy (kcal) and macronutrient (% of diet composition) intake calculated for all food consumed over 2 consecutive days (inpatient). | 17.6 | ▲ | ▼ | ▲ | ▼ |
| **Overall quality of quantitative non-randomised study evidence** | | | | | | | | ⊕○○○ | | | |
| **STUDIES OF HEALTHY VOLUNTEERS** | | | | | | | | | | | |
| High | (Daurignac et al., 2015) | 2-arm double-blind parallel RCT | 2 weeks | 13 | Olanzapine | Protein intake (g) calculated for all food consumed from a standardised lab breakfast meal (lab). | NA | | | | ▲ |
| Normal | (Ballon et al., 2018) | 3-arm double-blind, parallel RCT | 28 days | 7 | Olanzapine | Energy (kcal) and macronutrient intake (g) calculated for all food consumed from a standardised lab lunch meal (lab). | NA | ▲ | ▲ | ▲ | ▲ |
| | (Ballon et al., 2018) | 3-arm double-blind, parallel RCT | 28 days | 7 | Iloperidone | Energy (kcal) calculated for all food consumed from a standardised lab lunch meal (lab). | NA | ▼ | | | |
| | (Roerig et al., 2005) | 3-arm parallel, double-blind RCT | 2 weeks | 16 | 0lanzapine | Energy intake (kcal) calculated for all food consumed at the dinner session (lab). | NA | ▲ | | | |
| | (Roerig et al., 2005) | 3-arm parallel, double-blind RCT | 2 weeks | 16 | Risperidone | Energy intake (kcal) calculated for all food consumed at the dinner session (lab). | NA | ▼ | | | |
| | (Teff et al., 2015) | 3-arm parallel, double-blind RCT | 12 days | 10 | Olanzapine | Energy intake (kcal) calculated for ad libitum food intake (lab). | NA | ▲ | | | |
| | (Teff et al., 2015) | 3-arm parallel, double-blind RCT | 12 days | 10 | Aripiprazole | Energy intake (kcal) calculated for ad libitum food intake (lab). | NA | ▼ | | | |
| **Overall quality of evidence from studies of healthy volunteers** | | | | | | | | ⊕○○○ | | | |

**Fig 4. Within-group syntheses - Effect direction plot of antipsychotics on food intake and dietary composition.** BMI = body mass index; g = grams; kcal = kilocalories; n = number; NA = not applicable; RCT = randomised controlled trial. Effect direction: upward arrow ▲ = positive effect; downward arrow ▼ = negative effect; sideways arrow ◄► = no change/mixed effect. Sample size: large arrow ▲ ≥ 300; medium arrow ▲ ≥ 100; small arrow ▲ 50-99; very small arrow ▲ < 50. Overall quality of evidence: ⊕○○○ very-low quality.

| BMI | Citation | Study design | End of study assessment | Sample size (n) | Antipsychotic | Measurement scale | Washout period (days) | Overall scores | Dietary restraint | Disinhibition | Hunger | Uncontrolled eating | Emotional eating |
|---|---|---|---|---|---|---|---|---|---|---|---|---|---|
| **QUANTITATIVE RCT EVIDENCE** | | | | | | | | | | | | | |
| High | Case et al., 2010 Study 3 (Hoffman et al., 2009) | 3-arm parallel, open label RCT | 2 weeks | 17 | Olanzapine | TEFQ | NR | | ▲ | ▼ | ▼ | | |
| Normal | (Kang et al., 2024) | 2-arm parallel, double-blind RCT | 5 days | 19 | Olanzapine | TFEQ-R21 | No (no previous APs) | ▼ | ▼ | | | ▼ | ▼ |
| **Overall quality of RCT evidence** | | | | | | | | | ⊕○○○ | ⊕○○○ | ⊕○○○ | | |
| **QUANTITATIVE NON-RANDOMISED STUDY EVIDENCE** | | | | | | | | | | | | | |
| High | (Stip et al., 2012) | Non-randomised controlled trial | 16 weeks | 15 | Olanzapine | TFEQ | No | | ▼ | ◄► | ▲ | | |
| **Overall quality of quantitative non-randomised study evidence** | | | | | | | | | ⊕○○○ | ⊕○○○ | ⊕○○○ | | |
| **STUDIES OF HEALTHY VOLUNTEERS** | | | | | | | | | | | | | |
| High | (Mathews et al., 2012) | Pre-post study | 7 days | 19 | Olanzapine | TEFQ | NA | ▲ | | ▲ | | | |
| **Overall quality of evidence from studies of healthy volunteers** | | | | | | | | | | ⊕○○○ | | | |

**Fig 5. Within-group syntheses - Effect direction plot of antipsychotics on eating cognitions and behaviours.** BMI = body mass index; n = number; NA = not applicable; NR = not reported; TFEQ = Three-Factor Eating Questionnaire; TFEQ-R21 = Three-Factor Eating Questionnaire-Revised 21-item Version; RCT = randomised controlled trial. Effect direction: upward arrow ▲ = positive effect; downward arrow ▼ = negative effect; sideways arrow ◄► = no change/mixed effect. Sample size: large arrow ▲ ≥ 300; medium arrow ▲ ≥ 100; small arrow ▲ 50-99; very small arrow ▲ < 50. Overall quality of evidence: ⊕○○○ very-low quality.

with the exception of dietary restraint. Patients with high baseline BMI [95] showed an increase in dietary restraint scores, while antipsychotic-naïve patients with normal baseline BMI [86] showed a decrease in the scores of this domain.

A non-randomised trial [73] studied the effect of a 16-week olanzapine treatment on Three-Factor Eating Questionnaire scores in overweight patients and showed a negative effect on restraint, positive effect on susceptibility to hunger but no effect on disinhibition.

Using GRADE, there was very low-level RCT evidence that antipsychotics (olanzapine) had an inconsistent effect on dietary restraint, decreased disinhibition and hunger scores; and very low-level non-randomised trial evidence that antipsychotics (olanzapine) decreased dietary restraint, had no effect on disinhibition and increased hunger scores.

A pre-post study of healthy volunteers with high baseline BMI [74] investigated the effect of a 7-day olanzapine treatment on Three-Factor Eating Questionnaire scores and found an increase in the overall and disinhibition scores from baseline. Using GRADE, there was very low-level evidence that olanzapine increased disinhibition scores among healthy volunteers with high baseline BMI.

The GRADE Summary of Findings Table for key outcomes for within-group syntheses is presented in Fig 6.

#### 4.3.2. Results of between-group syntheses.

a) Appetite sensations

Fig 7 shows the effect direction plot for this section of the results.

*Appetite as a composite outcome:* Only one randomised placebo-controlled trial [70] of healthy volunteers with normal baseline examined the effect of olanzapine and risperidone on

| Question: What are the effects of antipsychotic medications on key eating-related outcomes in within-group syntheses? | | | | |
|---|---|---|---|---|
| | | | **Summary of findings** | |
| **Outcome** | **Study design** | **No. of participants (studies)** | **Vote counting with direction of effect** | **Quality** |
| **Appetite** | RCT (Hardy et al., 2009; Karagianis et al., 2009; Park et al., 2013; Smith et al., 2012) | 219 (4) | 3 of 4 samples with high baseline BMI (reported in 3 studies) receiving olanzapine or risperidone reported decline in appetite from baseline, 1 reported increase in appetite. 2 samples with normal baseline BMI (reported in 1 study) receiving ziprasidone or olanzapine reported no effect. | Very low ⊕○○○ |
| | Prospective, observational (Treuer et al., 2009) | 606 (1) | 1 prospective observational study including olanzapine-treated patients with normal baseline reported a decline in appetite scores from baseline. | Very low ⊕○○○ |
| **Hunger** | RCTs of healthy volunteers (Ballon et al., 2018; Roerig et al., 2005; Teff et al., 2013) | 50 (3) | The impact of olanzapine, iloperidone and aripiprazole on hunger was assessed in a laboratory setting in 3 RCTs of healthy volunteers with normal baseline BMI. They all reported an increase in hunger scores from baseline. | Very low ⊕○○○ |
| **General food cravings** | RCT (Hoffman et al., 2009) | 50 (1) | 1 RCT including participants with high baseline BMI reported a decline in general food craving scores from baseline | Low ⊕⊕○○ |
| | Prospective, observational (Garriga et al., 2019) | 34 (1) | 1 prospective, observational study including 2 independent samples stratified by baseline BMI, investigated the effect of an 18-week clozapine treatment on food cravings. Normal weight patients experienced a consistent increase in general food cravings from baseline, whereas overweight or obese patients experienced a decline. | Very low ⊕○○○ |
| **Energy intake** | Prospective observational study (Gothelf et al., 2002) | 10 (1) | 1 prospective observational study examined the change in energy intake among olanzapine-treated patients with normal baseline BMI. It reported an increase in energy intake from baseline. | Very low ⊕○○○ |
| | RCTs of healthy volunteers (Ballon et al., 2018; Roerig et al., 2005; Teff et al., 2013) | 66 (3) | 3 RCTs of healthy volunteers with normal baseline BMI reported an increase in energy intake from baseline in olanzapine-treated participants. A contradictory effect was observed in healthy volunteers receiving iloperidone, risperidone and aripiprazole. | Very low ⊕○○○ |
| **Eating cognitions: restraint** | RCT (Hoffman et al., 2009; Kang et al., 2024) | 36 (2) | 1 RCT of patients with high BMI showed an increase in restraint scores over a 2-week olanzapine treatment, while 1 RCT of AP-naïve patients with normal BMI showed a decrease in restraint scores over a 5-day olanzapine treatment. | Very low ⊕○○○ |
| | Non-randomised controlled trial (Stip et al., 2012) | 15 (1) | 1 non-randomised trial of overweight patients studied the effect of a 16-week olanzapine treatment and showed a negative effect on restraint. | Very low ⊕○○○ |
| **Eating cognitions: disinhibition** | RCT (Hoffman et al., 2009) | 17 (1) | 1 RCT of patients with high baseline BMI showed a decline in disinhibition scores over a 2-week olanzapine treatment. | Very low ⊕○○○ |
| | Non-randomised controlled trial (Stip et al., 2012) | 15 (1) | 1 non-randomised trial of overweight patients studied the effect of a 16-week olanzapine treatment and showed no effect on disinhibition. | Very low ⊕○○○ |
| | Pre-post study of healthy volunteers (Mathews et al., 2012) | 19 (1) | 1 pre-post study of healthy volunteers with high baseline BMI investigated the effect of a 7-day olanzapine treatment on TFEQ scores and found an increase in disinhibition scores from baseline. | Very low ⊕○○○ |
| **Eating cognitions: hunger** | RCT (Hoffman et al., 2009) | 17 (1) | 1 RCT of patients with high baseline BMI reported a decline in hunger scores over a 2-week olanzapine treatment. | Very low ⊕○○○ |
| | Non-randomised controlled trial (Stip et al., 2012) | 15 (1) | 1 non-randomised trial of overweight patients studied the effect of a 16-week olanzapine treatment on TEFQ scores and reported an increase in hunger scores. | Very low ⊕○○○ |

**Fig 6. Within-group syntheses GRADE Summary of Findings Table.** BMI = body mass index; n = number; RCT = randomised controlled trial; TFEQ = Three-Factor Eating Questionnaire.

appetite. It reported higher appetite scores in the olanzapine and risperidone arms compared to placebo over a 2-week period. Using GRADE, there was very low-level RCT evidence that among healthy volunteers with normal baseline BMI, antipsychotic-treated participants had higher appetite scores than controls.

*Hunger, satiety and fullness:* The effect of second-generation antipsychotics on hunger and satiety were investigated in a laboratory setting in one cross-sectional study [27]. Second-generation antipsychotic-treated patients with high baseline BMI had higher hunger scores compared to the unexposed healthy participants. However, the effect on satiety, measured as the satiety quotient after standardised meals, was inconsistent. Using GRADE, there was very low-level observational study evidence that second-generation antipsychotic-treated patients with high baseline BMI had higher hunger scores compared to unexposed healthy participants.

Two randomised placebo-controlled trials of healthy volunteers with normal baseline BMI [65,99] measured hunger using a visual analogue scale in a laboratory setting and found higher pre-meal [65] and total [99] hunger scores in the olanzapine arm compared to placebo. A contradictory effect was reported in the iloperidone [65] and aripiprazole arms [99]. The latter RCT also found lower total fullness scores [99] in the olanzapine arm compared to placebo. Using GRADE, there was low-level RCT evidence that among healthy volunteers with normal baseline BMI, olanzapine-treated participants had higher hunger scores compared to controls.

| BMI | Citation | Study design | End of study assessment | Sample size (n) | Antipsychotic | Measurement scale | Appetite (composite) | Cravings | | | | | Hunger | SQ after standardised meals | Fullness |
|---|---|---|---|---|---|---|---|---|---|---|---|---|---|---|---|
| | | | | | | | | General food cravings | Carbohydrates | Sweets | Fast-food fats (FFF) | High fats | | | |
| *QUANTITATIVE NON-RANDOMISED STUDY EVIDENCE* | | | | | | | | | | | | | | | |
| High | (Abbas and Liddle, 2013) | Cross-sectional | NA | Olanzapine-treated patients/unexposed group n= 20/20 | Olanzapine | FCI | | ▲ | ▲ | ▼ | ▼ | ▲ | | | |
| | (Abbas and Liddle, 2013) | Cross-sectional | NA | FGA-treated patients/ unexposed group n= 20/20 | FGA | FCI | | ▲ | ▲ | ▲ | ▲ | ▲ | | | |
| | (Blouin et al., 2008) | Cross-sectional | NA | SGA-treated patients/ Unexposed group n= 18/20 | SGA | VAS [a] | | | | | | | ▲ | ◄► | |
| **Overall quality of quantitative non-randomised study evidence** | | | | | | | | ⊕○○○ | | | | | | ⊕○○○ | |
| *STUDIES OF HEALTHY VOLUNTEERS* | | | | | | | | | | | | | | | |
| Normal | (Ballon et al., 2018) | 3-arm double-blind, parallel RCT | 28 days | Olanzapine/placebo n= 7/10 | Olanzapine | VAS [a] | | | | | | | ▲ | | |
| | (Ballon et al., 2018) | 3-arm double-blind, parallel RCT | 28 days | Iloperidone/placebo n= 7/10 | Iloperidone | VAS [a] | | | | | | | ▼ | | |
| | (Roerig et al., 2005) | 3-arm parallel, double-blind RCT | 2 weeks | Olanzapine/placebo n= 16/16 | Olanzapine | VAS [a] | ▲ | | | | | | | | |
| | (Roerig et al., 2005) | 3-arm parallel, double-blind RCT | 2 weeks | Risperidone/placebo n= 16/16 | Risperidone | VAS [a] | ▲ | | | | | | | | |
| | (Teff et al., 2015) | 3-arm parallel, double-blind RCT | 12 days | Olanzapine/placebo n=10/10 | Olanzapine | VAS [a] | | | | | | | ▲ | | ▼ |
| | (Teff et al., 2015) | 3-arm parallel, double-blind RCT | 12 days | Aripiprazole/placebo n=10/10 | Aripiprazole | VAS [a] | | | | | | | ▼ | | |
| **Overall quality of evidence from studies of healthy volunteers** | | | | | | | | ⊕○○○ | | | | | | ⊕⊕○○ | |

**Fig 7. Between-group syntheses - Effect direction plot of antipsychotics on appetite sensations. a = unvalidated measurement scale.** BMI = body mass index; FCI = Food Craving Inventory; FGA = first-generation antipsychotic; n = number; NA = not applicable; RCT = randomised controlled trial; SQ = satiety quotient; SGA = second-generation antipsychotic; VAS = Visual analogue scale. Effect direction: upward arrow ▲ = positive effect; downward arrow ▼ = negative effect; sideways arrow ◄► = no change/mixed effect. Sample size: large arrow ▲ ≥ 300; medium arrow ▲ ≥ 100; small arrow ▲ 50-99; very small arrow ▲ < 50. Overall quality of evidence: ⊕○○○ very-low quality; ⊕⊕○○ low quality.

**Food cravings:** Food cravings were assessed in one cross-sectional study [54] using the Food Craving Inventory (FCI). It found higher general food, carbohydrates and high fats cravings among olanzapine- and first-generation antipsychotic-treated patients with high baseline BMI compared to unexposed healthy participants. However, compared to the unexposed group, the olanzapine-treated patients had lower scores on specific cravings for sweets and fast-food fats (FFF), while the first-generation antipsychotic-treated patients had higher scores. Using GRADE, there was very low-level observational study evidence that antipsychotic-treated (olanzapine, first-generation antipsychotics) patients with high baseline BMI had higher general food cravings scores compared to unexposed healthy participants.

b) Food intake and dietary composition:

Fig 8 shows the effect direction plot for this section of the results.

Eight cross-sectional studies [27,53,56,58,59,75,77,91], including 11 independent samples, examined the effect of antipsychotics on daily energy and macronutrient intake and showed inconsistent results.

Antipsychotics had an inconsistent effect on total energy intake in patients with high baseline BMI [27,56,58,59,77,91] and normal BMI [53], compared to unexposed healthy participants. Antipsychotics had a positive effect on total fat intake in antipsychotic-treated patients with high baseline BMI [58,59,75] (n = 4/7 samples) and an inconsistent effect in those with normal BMI [53]. A breakdown of fat consumption showed a positive effect of antipsychotics on saturated fatty acid intake in antipsychotic-treated patients with high BMI [58,59] (n =

| BMI | Citation | Study design | End of study assessment | Sample size (n) | Antipsychotic | Measurement scale | Energy | Fruit & Vegetable | Total fat | SFA | MUFA | PUFA | Cholesterol | Total carbohydrates | Starch | Soluble carbohydrates | Proteins | Fibre | Alcohol | Unhealthy diet |
|---|---|---|---|---|---|---|---|---|---|---|---|---|---|---|---|---|---|---|---|---|
| **QUANTITATIVE NON-RANDOMISED STUDY EVIDENCE** | | | | | | | | | | | | | | | | | | | | |
| High | (Archie et al., 2007) | Cross-sectional | NA | AP-treated patients/Unexposed group n=101/208 | FGA, SGA | Fruit and Vegetable and Fibre Screener; Dietary Fat Screener | | ▲ | ▲ | | | | | | | | | | | |
| | (Blouin et al., 2008) | Cross-sectional | NA | SGA-treated patients/Unexposed group n= 18/20 | SGA | Food weighed before and after ad libitum buffet meal (lab). | ▼ | | ▼ | | | | | ▼ | | | ▼ | | | |
| | (Henderson et al., 2006) | Cross-sectional | NA | SGA-treated patients/Unexposed group n= 88/723 | SGA | SGA-treated group: self-rated 4-day dietary record, FFQ Unexposed group 24-hour dietary recall method | ▼ | | ▼ | ▼ | ▼ | ▼ | ▼ | | | | ▼ | ▼ | ▼ | |
| | (Jakobsen et al., 2018b) | Cross-sectional | NA | SGA-treated patients/ Unexposed group: n=346/3016 | SGA | AP-treated group: 24-hr recall. Unexposed group: 7-day food record. | ▼ | ▼ | | | | | | | | | | ▼ | ▲ | ▲ |
| | (Nunes et al., 2014) | Cross-sectional | NA | AP-treated patients/Unexposed group n= 25/25 | FGA, SGA | FFQ | ▲ | | | | | | | | | | | | | |
| | (Stefańska et al., 2017) | Cross-sectional | NA | Female AP-treated/female unexposed: n= 32/60 | FGA, SGA | 24-hr recall | ▲ | | ▲ | ▲ | ▲ | ▲ | ▼ | ▲ | | | ▲ | ▲ | | |
| | (Stefańska et al., 2017) | Cross-sectional | NA | Male AP-treated/male unexposed n= 28/38 | FGA, SGA | | ▲ | | ▲ | ▲ | ▲ | ▲ | ▲ | ▲ | | | ▼ | ▼ | | |
| | (Stefańska et al., 2018) | Cross-sectional | NA | Female AP-treated/ female unexposed: n=45/40 | FGA, SGA | 24-hr recall | ▲ | | ▲ | ▲ | ▲ | ▼ | ▼ | ▲ | | | ▼ | ▼ | | |
| | (Stefańska et al., 2018) | Cross-sectional | NA | Male AP-treated/male unexposed n=40/30 | FGA, SGA | | ▼ | | ▼ | ▲ | ▼ | ▼ | ▼ | ▼ | | | ▼ | ▼ | | |
| Normal | (Saugo et al., 2020) | Cross-sectional | NA | Female AP-treated/ female unexposed: n=21/1245 | FGA, SGA | AP-treated group: EPIC Questionnaire Unexposed group: self-recorded food consumption for 3 days | ▲ | | ▲ | ▲ | ▼ | ▲ | ▲ | ▲ | ▼ | ▲ | ▲ | ▲ | ▲ | |
| | (Saugo et al., 2020) | Cross-sectional | NA | Male AP-treated/male unexposed n=33/1068 | FGA, SGA | | ▼ | | ▼ | ▲ | ▼ | ▼ | ▲ | ▲ | ▼ | ▲ | ▲ | ▲ | ▼ | |
| **Overall quality of quantitative non-randomised study evidence** | | | | | | | ⊕○○○ | | | | | | | | | | | | | |
| **STUDIES OF HEALTHY VOLUNTEERS** | | | | | | | | | | | | | | | | | | | | |
| Normal | (Fountaine et al., 2010) | Randomised, double blind, placebo controlled 2- period crossover trial | Each period: 15 days; 12-day washout | n = 30 (21 completers) | Olanzapine | Energy intake (kcal) calculated for all food consumed on day 14 of study periods 1 and 2 (lab). | ▲ | | | | | | | | | | | | | |
| | (Teff et al., 2013) | 3-arm parallel, double-blind RCT | 12 days | Olanzapine/placebo n=10/10 | Olanzapine | Energy intake (kcal) calculated for ad libitum food intake (lab). | ▲ | | | | | | | | | | | | | |
| | (Teff et al., 2013) | 3-arm parallel, double-blind RCT | | Aripiprazole/ placebo n=10/10 | Aripiprazole | | ▼ | | | | | | | | | | | | | |
| **Overall quality of evidence from studies of healthy volunteers** | | | | | | | ⊕○○○ | | | | | | | | | | | | | |

**Fig 8. Between-group syntheses - Effect direction plot of antipsychotics on food intake and dietary composition.** AP = antipsychotic; BMI = body mass index; EPIC = European Prospective Investigation into Cancer and Nutrition Questionnaire; FFQ = food frequency questionnaire; FGA = first-generation antipsychotic; kcal = kilocalories; MUFA = monounsaturated fatty acids; n = number; NA = not applicable; PUFA = polyunsaturated fatty acids; RCT = randomised controlled trial; SFA = saturated fatty acids; SGA = second-generation antipsychotic. Effect direction: upward arrow ▲ = positive effect; downward arrow ▼ = negative effect; sideways arrow ◄► = no change/mixed effect. Sample size: large arrow ▲ ≥ 300; medium arrow ▲ ≥ 100; small arrow ▲ 50-99; very small arrow ▲ < 50. Overall quality of evidence: ⊕○○○ very-low quality.

4/5) and normal BMI [53] (n = 2/2). Antipsychotics had a positive effect on monounsaturated fatty acid (MUFA) intake in antipsychotic-treated patients with high baseline BMI [58,59] (n = 3/5), but a negative effect in those with normal BMI [53] (n = 2/2). Antipsychotics had a negative effect on polyunsaturated fatty acid (PUFA) intake in antipsychotic-treated patients with high BMI [59,77] (n = 3/5), but an inconsistent effect in those with normal BMI [53]. Antipsychotics had a negative effect on cholesterol intake in patients with high baseline BMI [58,59,77] (n = 4/5), but had a positive effect in those with normal BMI [53] (n = 2/2).

Antipsychotics had an inconsistent effect on total carbohydrate intake in antipsychotic-treated patients with high BMI [27,58,59,77], but a positive effect in those with normal BMI [53] (n = 2/2). A breakdown of carbohydrate consumption showed that antipsychotics had a positive effect on soluble carbohydrate intake (n = 2/2) and a negative effect on starch intake in antipsychotic-treated patients with normal BMI [53] (n = 2/2). Antipsychotics had a mixed effect on fruit and vegetable intake in antipsychotic-treated patients with high BMI [56,75];

they had an inconsistent effect on alcohol intake in patients with high [56,77] and normal BMI [53].

Using GRADE, there was very low-level observational study evidence that antipsychotics had an inconsistent effect on daily energy intake.

Two randomised placebo-controlled trials of healthy volunteers with normal baseline BMI concluded that olanzapine-treated participants (for 15 and 12 days, respectively) consumed more calories [68,72], while aripiprazole-treated participants consumed less calories [72] than those in the placebo arm. Using GRADE, there was very low-level RCT evidence that olanzapine-treated healthy volunteers with normal baseline BMI had higher energy intake, while aripiprazole-treated participants had lower energy intake, compared to placebo.

c) Eating disorders:

Fig 9 shows the effect direction plot for this section of the results.

Only 2 cross-sectional studies [63,81] investigated the association between antipsychotics and eating disorders. Second-generation antipsychotic-treated patients had higher odds of developing binge eating disorders (BED) compared to unexposed healthy participants [63], regardless of baseline BMI. Using GRADE, there was very low-level observational study evidence that second-generation antipsychotic-treated patients were more likely to develop binge eating disorders than unexposed healthy participants, regardless of baseline BMI.

Disordered eating behaviours, assessed using the Eating Attitude Test-26, were more likely in antipsychotic-treated patients compared to unexposed healthy participants [81]. However, one study [63] assessed the effect of BMI, using a *post hoc* threshold, on developing binge symptoms (i.e., binge episodes less than two days/week). Among participants with BMI lower than 28 (mean BMI = 23.6; SD = 2.2), the odds of binge symptoms were lower in second-generation antipsychotic-treated patients than healthy participants, while a contradictory effect was observed among those with BMI 28 or over (mean BMI = 32.9; SD = 6.1).

| BMI | Citation | Study design | End of study assessment | Sample size (n) | Antipsychotic | Measurement scale | Eating disorder | | |
|---|---|---|---|---|---|---|---|---|---|
| | | | | | | | Binge symptoms (binge episodes <2 days/week) | Disordered eating (EAT-26 total score ≥20) | Binge eating disorder (BED) |
| *QUANTITATIVE NON-RANDOMISED STUDY EVIDENCE* | | | | | | | | | |
| High | (Khazaal et al., 2006a) | Cross-sectional | NA | SGA-treated patients/unexposed, among those with BMI ≥28 [a] n= 20/20 | SGA | DSM-IV criteria | ▲ | | ▲ |
| Normal | (Khazaal et al., 2006a) | Cross-sectional | NA | SGA-treated patients/unexposed, among those with BMI < 28 [b] n= 20/20 | SGA | DSM-IV criteria | ▼ | | ▲ |
| NR | (Khosravi, 2020) | Cross-sectional | NA | AP-treated patients/Unexposed n= 154/154 | FGA, SGA | Total score of ≥20 on the EAT-26 | | ▲ | |
| Overall quality of quantitative non-randomised study evidence | | | | | | | | | ⊕○○○ |

**Fig 9. Between-group syntheses - Effect direction plot of antipsychotics on the odds of developing eating disorders. a: Mean (SD) BMI of SGA-treated group** mean = 32.90 (6.10), unexposed group = 33.80 (4.90). **b: Mean (SD) BMI of SGA-treated group** = 23.60 (2.20), unexposed group = 21.10 (2.50). AP = antipsychotic; BED = binge eating disorder; BMI = body mass index; DSM-IV = Diagnostic and Statistical Manual of Mental Disorders, 4th Edition; EAT-26 = Eating Attitude Test; FGA = first-generation antipsychotic; n = number; NA = not applicable; NR = not reported; RCT = randomised controlled trial; SGA = second-generation antipsychotic. Effect direction: upward arrow ▲ = positive effect; downward arrow ▼ = negative effect; sideways arrow ◀▶ = no change/mixed effect. Sample size: large arrow ▲ ≥ 300; medium arrow ▲ ≥ 100; small arrow ▲ 50-99; very small arrow ▲ < 50. Overall quality of evidence: ⊕○○○ very-low quality.

d) Eating cognitions and behaviours:

Fig 10 shows the effect direction plot for this section of the results.

Five cross-sectional studies [27–29,57,101], including eight independent samples, investigated the effect of antipsychotics on eating cognitions and behaviours using validated questionnaires. Two studies concluded that antipsychotic-treated patients had consistently higher scores for distorted cognitions compared to healthy participants, while controlling for the effect of weight [27,57].

Two studies assessed the effect of weight, using a *post hoc* threshold, and reported conflicting results. One study [101] found that, among participants with BMI 28 or over, those treated with second-generation antipsychotics had a lower overall score for distorted cognitions than their healthy counterparts, mainly due to lower 'self-control' scores; whereas patients with BMI less than 28 had higher scores than their healthy counterparts. On the other hand, a second study [28] concluded that second-generation antipsychotics increased distorted cognitions regardless of weight.

One study [29], including two independent samples, concluded that first-generation antipsychotic-treated participants had lower scores for distorted cognitions compared to healthy unexposed participants, while the effect among those treated with second-generation antipsychotics was unclear.

*Dietary restraint:* Dietary restraint was assessed using the Three-Factor Eating Questionnaire (TEFQ), Three-Factor Eating Questionnaire-Revised 21-Item Version (TEFQ-R21) or Revised version of the Mizes Anorectic Cognitions questionnaire (MAC-R). Five of six samples of antipsychotic-treated patients with high baseline BMI [27–29,57,101], and all (n = 2/2) samples of antipsychotic-treated patients with normal baseline BMI [28,101] reported higher

| BMI | Citation | Study design | End of study assessment | Sample size (n) | Antipsychotic | Measurement scale | Overall scores | Dietary restraint | Disinhibition | Hunger | Uncontrolled eating | Emotional eating | Appearance, weight, and approval | Self-Control of eating, self-esteem |
|---|---|---|---|---|---|---|---|---|---|---|---|---|---|---|
| | **QUANTITATIVE NON-RANDOMISED STUDY EVIDENCE** | | | | | | | | | | | | | |
| High | (Blouin et al., 2008) | Cross-sectional | NA | SGA-treated patients/Unexposed group n= 18/20 | SGA | TFEQ | | ▲ | ▲ | ▲ | | | | |
| | (Khazaal et al., 2006b) | Cross-sectional | NA | SGA-treated patients/Unexposed, among those with BMI ≥28 n= 20/20 [a] | SGA | MAC-R | ▼ | ▲ | | | | | ▲ | ▼ |
| | (Khazaal et al., 2009) | Cross-sectional | NA | SGA-treated patients/Unexposed, among those with weight gain (minimum 2 kg in the last month) n= 10/5 | SGA | TEFQ | ▲ | ▲ | ▲ | ▲ | | | | |
| | (Kouidrat et al., 2018) | Cross-sectional | NA | AP-treated patients/Unexposed n= 66/81 | FGA, SGA, both | TEFQ-R21 | | ▲ | | | ▲ | ▲ | | |
| | (Sentissi et al., 2009) | Cross-sectional | NA | SGA-treated patients/Untreated patients n= 93/33 | SGA | TEFQ/DEBQ | | ▲ TFEQ | ▼ TFEQ | ▼ TFEQ | ◄► DEBQ | ▲ DEBQ | | |
| | (Sentissi et al., 2009) | Cross-sectional | NA | FGA-treated patients /Untreated patients n= 27/33 | FGA | TEFQ/DEBQ | | ▼ TFEQ | ▼ TFEQ | ▼ TFEQ | ▼ DEBQ | ▼ DEBQ | | |
| Normal | (Khazaal et al., 2006b) | Cross-sectional | NA | SGA-treated patients /Unexposed group, among those with BMI <28 [b] n= 20/20 | SGA | MAC-R | ▲ | ▲ | | | | | ▲ | ▲ |
| | (Khazaal et al., 2009) | Cross-sectional | NA | SGA-treated patients/Unexposed, among those without weight gain (neither gained nor lost more than 1 kg in the last month) n= 12/10 | SGA | TEFQ | ▲ | ▲ | ▲ | ▲ | | | | |
| | **Overall quality of quantitative non-randomised study evidence** | | | | | | ⊕○○○ | ⊕○○○ | ⊕○○○ | | | | | |

**Fig 10. Between-group syntheses - Effect direction plot of antipsychotics on eating cognitions and behaviour.** a: Mean (SD) BMI of SGA-treated group mean = 32.90 (6.10), unexposed group = 33.80 (4.90). b: Mean (SD) BMI of SGA-treated group = 23.60 (2.20), unexposed group = 21.10 (2.50). AP = antipsychotic; BMI = body mass index; DEBQ = Dutch Eating Behavior Questionnaire; FGA = first-generation antipsychotic; MAC-R = Revised version of the Mizes Anorectic Cognitions questionnaire; n = number; NA = not applicable; SGA = second-generation antipsychotic; TFEQ = Three-Factor Eating Questionnaire; TFEQ-R21 = Three-Factor Eating Questionnaire-Revised 21-Item Version. Effect direction: upward arrow ▲ = positive effect; downward arrow ▼ = negative effect; sideways arrow ◄► = no change/mixed effect. Sample size: large arrow ▲ ≥ 300; medium arrow ▲ ≥ 100; small arrow ▲ 50-99; very small arrow ▲ < 50. Overall quality of evidence: ⊕○○○ very-low quality.

restraint scores than unexposed groups. Using GRADE, there was very low-level observational study evidence that antipsychotic-treated patients had higher restraint scores than unexposed healthy participants, regardless of BMI.

*Dietary disinhibition and hunger:* Dietary disinhibition and hunger were assessed using the Three-Factor Eating Questionnaire (TFEQ) in a total of five samples (four with high baseline BMI, one with normal BMI). Antipsychotics had an inconsistent effect on disinhibition and hunger scores in antipsychotic-treated patients with high baseline BMI [27–29], but had a positive effect in one sample of antipsychotic-treated patients with normal baseline BMI [28]. Using GRADE, there was very low-level observational study evidence that antipsychotics had an inconsistent effect on disinhibition and hunger scores among patients with high baseline BMI, while having a positive effect among those with normal baseline BMI.

Uncontrolled and emotional eating were measured in three samples with high baseline BMI [29,57] using the Dutch Eating Behaviour Questionnaire (DEBQ) and the Three-Factor Eating Questionnaire-Revised 21-Item Version (TEFQ-R21), respectively. They had an inconsistent effect on uncontrolled eating scores, but increased emotional eating scores (n = 2/3).

The GRADE Summary of Findings Table for key outcomes for between-group syntheses is presented in Fig 11.

| Question: What are the effects of antipsychotic medications on key eating-related outcomes in between-group syntheses? | | | | |
|---|---|---|---|---|
| | | | Summary of findings | |
| Outcome | Study design | No. of participants [exposed/unexposed] (studies) | Vote counting with direction of effect | Quality |
| Appetite | RCT of healthy volunteers (Roerig et al., 2005) | 48 [32/16] (1) | 1 RCT of healthy volunteers with normal baseline BMI reported higher appetite scores in the olanzapine and risperidone arms compared to placebo. | Very low ⊕○○○ |
| Hunger | Cross-sectional (Blouin et al., 2008) | 38 [18/20] (1) | 1 cross-sectional study found that SGA-treated patients with high baseline BMI had higher hunger scores compared to unexposed healthy participants. | Very low ⊕○○○ |
| | RCT of healthy volunteers (Ballon et al., 2018; Teff et al., 2015) | 54 [34/20] (2) | 2 RCTs of healthy volunteers with normal baseline found that olanzapine-treated participants had higher pre-meal and total hunger scores than participants in the placebo arm. A contradictory effect was reported in the iloperidone and aripiprazole arms. | Low ⊕⊕○○ |
| General food cravings | Cross-sectional (Abbas and Liddle, 2013) | 60 [40/20] (1) | 1 cross-sectional study found that olanzapine- and FGA-treated patients with high baseline BMI had higher general food cravings scores than unexposed healthy participants. | Very low ⊕○○○ |
| Energy intake | Cross-sectional (Blouin et al., 2008; Henderson et al., 2006; Jakobsen et al., 2018; Saugo et al., 2020; Stefańska et al., 2017; Stefańska et al., 2018) | 6941 [676/6265] (7) | Antipsychotics had an inconsistent effect on total energy intake in patients with high baseline BMI (total 8 samples reported in 6 studies) and normal BMI (2 samples reported in 1 study), compared to unexposed healthy participants. | Very low ⊕○○○ |
| | RCT of healthy volunteers (Fountaine et al., 2010; Teff et al., 2013) | 60 [50/10] (2) | 2 RCTs of healthy volunteers with normal baseline BMI found that olanzapine-treated participants consumed more calories, while aripiprazole-treated participants consumed less calories than those in the placebo arm. | Very low ⊕○○○ |
| Eating disorders: BED | Cross-sectional (Khazaal et al., 2006a) | 80 [40/40] (1) | 1 cross-sectional study found that SGA-treated patients had higher odds of developing binge eating disorders (BED) compared to unexposed healthy participants, regardless of baseline BMI. | Very low ⊕○○○ |
| Eating cognitions: restraint | Cross-sectional (Blouin et al., 2008; Khazaal et al., 2009; Khazaal et al., 2006b; Kouidrat et al., 2018; Sentissi et al., 2009) | 455 [266/189] (5) | 5 of 6 samples of antipsychotic-treated patients with high baseline BMI (reported in 5 studies), and all (n=2/2) samples of antipsychotic-treated patients with normal baseline BMI (reported in 2 studies) reported higher restraint scores than unexposed groups. | Very low ⊕○○○ |
| Eating cognitions: disinhibition | Cross-sectional (Blouin et al., 2008; Khazaal et al., 2009; Sentissi et al., 2009) | 228 [160/68] (3) | Antipsychotics had an inconsistent effect on disinhibition scores among patients with high baseline BMI (total 4 samples reported in 3 studies), while having a positive effect among those with normal baseline BMI (1 sample). | Very low ⊕○○○ |
| Eating cognitions: hunger | Cross-sectional (Blouin et al., 2008; Khazaal et al., 2009; Sentissi et al., 2009) | 228 [160/68] (3) | Antipsychotics had an inconsistent effect on hunger scores among patients with high baseline BMI (total 4 samples reported in 3 studies), while having a positive effect among those with normal baseline BMI (1 sample). | Very low ⊕○○○ |

**Fig 11. Between-group syntheses GRADE Summary of Findings Table.** BMI = body mass index; FGA = first-generation antipsychotic; n = number; RCT = randomised controlled trial; SGA = second-generation antipsychotic.

## 5. Results derived from qualitative studies

### 5.1. Overview of qualitative studies

Study and participant characteristics of the six included qualitative studies are presented in the S11 File.

The studies varied in terms of situational context, language used to conduct interviews, data collection and analysis methods. Two studies [102,103] were conducted in North America, two [104,105] in Australia, one [106] in the UK and one [107] in Ethiopia. The latter was the only study conducted in a setting with limited food availability. People with lived experience were not involved in the design, conduct or analysis of any of these studies.

One study [107] explored the perspectives of different stakeholders including patients, carers, mental health research field workers and a psychiatric nurse, while the remaining five studies [102–106] included patients' perspectives only. Three studies [104–106] recruited participants from community-based mental health services, one [103] from an Early Intervention Programme outpatient clinic, one [102] from a psychiatric outpatient clinic, and one [107] from a psychiatric nurse-led outpatient unit located in a rural setting.

Four studies [102–105] used individual interviews, one [106] used focus groups and think aloud sessions, and one [107] used a mixture of both individual interviews and focus groups. In the latter study [107] interviews were conducted in Amharic and transcripts were translated into English, but were not verified for their accuracy. Two studies [103,104] used grounded theory, two [106,107] used thematic analysis and one [105] used phenomenological analysis. One study [102] did not specify the analysis method used, but was believed to have used thematic analysis based on the description provided by the authors. The studies were published between 2012 and 2019.

The sample sizes ranged from 8 to 51 and included a total of 127 participants; 100 patients, 19 caregivers, seven mental health research field workers and one psychiatric nurse. Of the 100 patients, 67 had a diagnosis of schizophrenia, 10 schizoaffective disorder, 1 bipolar disorder, 11 first episode psychosis (FEP) and 11 were collectively described as having schizophrenia spectrum disorder (SSD). Their age ranged from 15 to 64 years. All patients received antipsychotic medications including first-generation antipsychotics [106,107], second-generation antipsychotics [102,103,105,106] or a combination of both [102,105,106]. One study [104] did not provide details of the antipsychotics used. One study [103] classified the participants based on duration of treatment.

### 5.2. Risk of bias assessment

Three studies [103–105] met all of the Mixed-Methods Appraisal Tool quality indicators. Five studies [102,103,105–107] failed to report on researcher reflexivity (the researchers' role was unclear or they were also healthcare providers directly involved with the treatment of patients). This was especially pertinent in one study [107] in which the interviewer was the head of the research project site, as well as the psychiatrist involved in treating patients and working with psychiatric nurses at the site. This may have potentially influenced what patients, carers and other healthcare professionals would report regarding their experiences. One study [104] explained how member checks were conducted in follow-up interviews to ensure the credibility of the findings. The studies mainly focused on experiences of antipsychotic-induced weight gain and its management [102–105], reasons for non-adherence to medications [107], and beliefs about antipsychotics and their role in patients' decision-making regarding antipsychotics [106]. Antipsychotic-induced changes in eating behaviours and appetite were explored in terms of their contribution to these phenomena; they were not the main focus of the included studies. Consequently, the

extracted qualitative evidence lacked conceptual richness. It is important to acknowledge that the validity of the resulting qualitative synthesis was impacted by the limited number of studies, lack of conceptual richness of the qualitative evidence and differences in situational context in terms of food security.

The risk of bias assessments of the 6 included qualitative studies is provided in the S12 File.

### 5.3.  Results of the qualitative synthesis

The findings suggest that across all studies most patients experienced changes in appetite following the use of antipsychotic medications. However, the relationship between antipsychotic medications and food intake was influenced by an interplay of individual, interpersonal and external factors, the most significant of which was the food environment.

Only one study was conducted in a setting [107] with limited food availability. Here, whether a patient's increased desire to eat was translated into increased food consumption was determined by their family's reactions to their increased appetite (interpersonal factor). Beliefs (individual factor) about managing side-effects of antipsychotic medication also played a role in patients' food consumption. In settings where food was available [102–106], a patient's increased desire to eat was translated into increased food consumption. Here, the effect of a patient's family was confined to perceived stigma. However, the quality of food consumed was determined by food affordability (external factor).

The qualitative evidence was categorised into four themes which were identified via inductive thematic analysis: (1) the experience of antipsychotic-induced changes in appetite and eating behaviours; (2) individual factor- beliefs about the role of food in negating side effects of antipsychotic medications; (3) interpersonal factor- family reactions; (4) external factor- food environment (availability, affordability). Differences between participants' experiences based on the availability and affordability of food are highlighted, where possible, across all themes. Participants' quotations are italicised and put in quotation marks; author quotations are put in quotation marks. The thematic framework is presented in Fig 12.

Using GRADE-CERQual, our assessment of the qualitative review findings ranged from low to very low. Summaries of each finding and their GRADE-CERQual assessment are provided in Table 1. Detailed description of our confidence assessments is provided in the GRADE-CERQual Evidence Profile (S13 File).

### Theme 1: The experience of antipsychotic-induced changes in appetite and eating behaviours

1a)  Range of experiences

Most participants in the included studies experienced an increase in appetite [102–107], hunger [103,106,107] and low satiety [103]. Increased appetite often referred to an increased desire to eat. Only one study mentioned that some participants experienced a decline in appetite [106]. In settings where food was available, an increase in appetite was translated into an increase in the frequency [103–106] and quantity [103,105] of food consumed. Participants believed that hunger led to 'continuous grazing', particularly of 'high-calorie' foods [103] and at night [103,106]. The overconsumption of food had immediate negative effects, *'I used to be bloated and (expletive) real sick from eating so much.'* [105]. Participants attributed these changes in eating behaviour to antipsychotic medications [102–107], but beliefs about how antipsychotic medications caused these changes were not explored in further detail. More importantly, participants were 'unprepared for the hunger they experienced' [103].

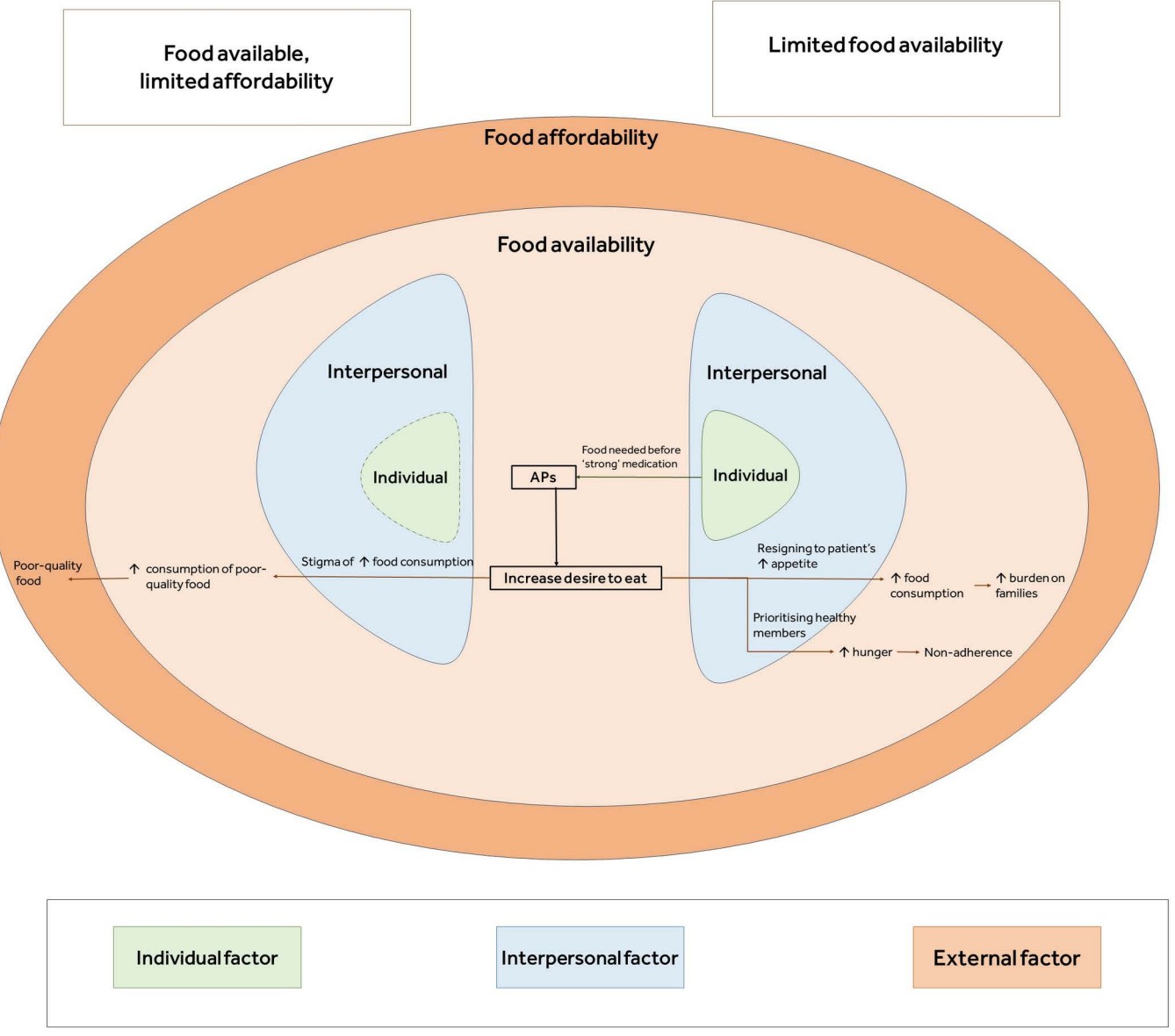

**Fig 12. Visual presentation of the thematic framework.** AP = antipsychotic medications.

**1b) Onset of experiences**

These experiences were believed to occur shortly after the introduction of antipsychotics [102,105].

> *'Instantly I noticed I started eating more...'* [105]

**1c) Intensity of experiences**

Strong word descriptors were used to portray the 'overwhelming intensity' [104] of these sensations: 'grappling with the desire to eat' [105], *'disastrously hungry'* [104], *'couldn't tolerate the hunger'* [107], having a *'bottomless pit'* stomach [103].

## Theme 2: Individual factor- beliefs about the role of food in negating side effects of antipsychotic medications

In the setting with limited food availability [107], participants believed that consuming *'good'* food was necessary to mitigate the adverse effects of *'strong'* antipsychotic medications which rendered their *'bodies are getting weaker'*. This increased the likelihood of non-adherence among patients who were already burdened with food shortages.

**Table 1. GRADE-CERQual summary of qualitative findings.**

| Objective: To synthesise qualitative evidence on the effect of antipsychotic medications on eating behaviours. Perspective: Experiences of people using antipsychotic medications on the effect of these medications on their eating beahviours; perspectives of carers/family members supporting people prescribed antipsychotic medications on the effect of these medications on eating behaviour. | | | |
|---|---|---|---|
| **Summary of review finding** | **Citations** | **GRADE-CERQual assessment of confidence in the evidence** | **Explanation of GRADE-CERQual assessment** |
| **Range of experiences:** Most participants experienced an increase in appetite, hunger and low satiety. Only one study mentioned that some participants experienced a decline in appetite. In settings where food was available, an increase in appetite was translated into an increase in the frequency and quantity of food consumed. Participants attributed changes in eating behaviour to antipsychotic medications and reported that they were *'unprepared for the hunger they experienced'*. | [102–107] | Very low | Moderate concerns regarding methodological limitations and adequacy of data; minor concerns regarding coherence and relevance of data. |
| **Onset of experiences:** These experiences were believed to occur shortly after the introduction of antipsychotics. | [102,105] | Very low | Moderate concerns regarding methodological limitations and adequacy of data; minor concerns regarding coherence and relevance of data. |
| **Intensity of experiences:** Strong word descriptors were used to portray the intensity of increased appetite and hunger. | [103–105,107] | Low | Moderate concerns regarding adequacy of data; minor concerns regarding methodological limitations, coherence and relevance of data. |
| **Individual factor- beliefs about role of food in negating side effects of antipsychotic medications:** In the setting with limited food availability, participants believed that consuming *'good'* food was necessary prior to taking the *'strong'* antipsychotic medications that rendered their bodies weak. | [107] | Very low | Serious concerns regarding adequacy of data; minor concerns regarding methodological limitations and relevance of data; no concerns regarding coherence of data. |
| **Interpersonal factor- family reactions:** In the setting with limited food availability, an increase in a patient's appetite resulted in one of two outcomes based on their family's reaction. Patients were denied access to more food when their families prioritised food access for healthy members who could secure livelihoods. Those patients experienced intolerable hunger which motivated them and their families to stop the medications. Families that resigned to their family member's increased desire to eat despite limited food availability, were burdened by their decision. In settings where food was available, an increase in appetite was translated into an increase in food consumption. The reactions of a patient's family to their increased appetite and food consumption was confined to perceived stigma. | [105,107] | Very low | Serious concerns regarding coherence and adequacy of data; minor concerns regarding methodological limitations and relevance of data. |
| **External factor - Food environment:** In settings where food was available, participants described how food prices (limited affordability) prohibited them from buying good quality food. In settings with limited food availability, participants described how food shortages (limited availability) interacted with family reactions and priorities to determine whether increased appetite was translated into increased food intake. | [102,103,107] | Very low | Serious concerns regarding adequacy of data; moderate concerns regarding coherence of data; minor concerns regarding methodological limitations and relevance of data. |

GRADE-CERQual = Grading of Recommendations, Assessment, Development, and Evaluations framework for Confidence in Evidence from Reviews of Qualitative research.

### Theme 3: Interpersonal factor- family reactions

In the setting with limited food availability [107], an increase in a patient's appetite resulted in one of two outcomes based on their family's reaction. Patients were denied access to more food when their families prioritised food access for healthy members who could secure livelihoods. Those patients experienced intolerable hunger which motivated them and their families to stop the medications. Families who resigned to their family member's increased desire to eat despite limited food availability, were burdened by their decision.

In settings where food was available, an increase in appetite was translated into increased food consumption [103,105]. Here, reactions of a patient's family to their increased appetite and food consumption was confined to perceived stigma. Patients tried to conceal the amount of food they consumed, *'...I used to sneak food... come out and sneak food out of the cupboard and hoard in my room...it was a big shame for my family to see me eating all the time.'* [105]

### Theme 4: External factor- Food environment

The included studies explored two aspects of food environment: food availability and affordability.

In settings where food was available, participants described how food prices (limited affordability) prevented them from buying good quality food, causing them to rely on unhealthy convenience foods instead. [102,103].

*'Sometimes it's just cheaper to buy something pre-made, and bad for you than it is to buy, you know, a couple pounds of vegetables and a sack of rice, or something to make, you know, a proper meal.'* [103]

In settings with limited food availability [107], participants described how food shortages (limited availability) interacted with family reactions and priorities to determine whether increased appetite was translated into increased food intake (discussed earlier).

## 6. Discussion

The aim of this review was to synthesise the available literature on the effect of antipsychotics on eating-related outcomes. Our search retrieved 61 studies; 55 quantitative and 6 qualitative studies, which were grouped by type of synthesis (i.e., within-group or between-group differences), core eating-related outcome domain, study design and type of participants. The main findings are discussed below, grouped by type of synthesis, followed by a discussion of the limitations of the evidence. The qualitative evidence is discussed in conjunction with the quantitative within-group comparisons owing to its focus on personal experiences of patients using antipsychotics.

### 6.1. Within-group comparisons

**6.1.1. Summary of evidence.** Overall, there was very low-level RCT evidence that olanzapine and risperidone decreased appetite in patients with high baseline BMI; very low-level RCT evidence that olanzapine and ziprasidone had an inconsistent effect on appetite in patients with normal baseline BMI; and very low-level observational study evidence that olanzapine decreased appetite in patients with normal baseline BMI. There was low-level RCT evidence (olanzapine) and very low-level observational study evidence (clozapine) that antipsychotics decreased general food cravings among patients with high baseline BMI. Observational study evidence showed that antipsychotics increased cravings among patients with normal baseline BMI. There was very low-level observational study evidence that olanzapine increased energy intake among patients with normal baseline BMI. There was very low-level RCT evidence that olanzapine had an inconsistent effect on dietary restraint,

decreased disinhibition and hunger scores; and very low-level non-randomised controlled trial evidence that olanzapine decreased dietary restraint, had no effect on disinhibition and increased hunger scores.

There was very low-level RCT and observational study evidence that antipsychotics increased hunger (olanzapine, iloperidone, aripiprazole), energy intake (olanzapine) and disinhibition scores (olanzapine) in healthy volunteers.

**6.1.2. Interpretation and limitations of the evidence.** Before interpreting the evidence, it is critical to remind the reader that within-group comparisons can be misleading. Within-group analyses address whether there is a change from baseline in a specific group, but cannot show whether change is greater in one group than the other. Moreover, changes from baseline can be due to natural changes over time or regression to the mean, rather than the actual effect of a treatment [108].

Evidence from within-group comparisons implies a potential inverse relationship between baseline BMI and the effect of antipsychotics on eating-related outcomes. However, this potential inverse relationship can be attributed to other factors, such as the type of participants included in the syntheses and the regression to the mean phenomenon. Most participants were chronic patients with high baseline BMI [52,66,71,80,94,95] who had received antipsychotic treatment prior to study entry. Thus, it is possible to assume that overweight/obese participants, who had already gained weight over the course of long-term antipsychotic treatment, had reached the proposed 'ceiling effect' of antipsychotics on induced weight gain [12,109,110]. Reaching this plateau weight could well be associated with a relative decline in appetite sensations and food intake. Moreover, none of the included studies provided sufficient detail on the intensity (dose and duration) of prior antipsychotic treatment. The absence of information about past treatment in studies involving short-term administration of particular antipsychotics makes it challenging to separate the effects of withdrawing previous antipsychotics from those of administering new ones.

Previous studies have identified a similar moderating effect of baseline BMI on the magnitude of antipsychotic-induced weight gain [111–113]. However, an analysis of datasets of seven long-term, randomised controlled trials of antipsychotic agents concluded that baseline weight/BMI was not a moderator of antipsychotic-induced weight gain [114]. Instead, regression to the mean was implicated as causing this spurious moderating effect. Regression to the mean is a common statistical artefact in repeated-measures analyses [115]. It occurs because unusually large, chance errors that contribute to initial extreme measurements (randomly large errors that tend to create very high or low data points) are unlikely to be repeated and tend to be followed by measurements that are closer to the true mean [116]. This effect of chance variation means that participants with relatively large values at recruitment for variable parameters (e.g., BMI or appetite) are likely to record less extreme values (lower BMI or appetite) at follow-up. In the within-group syntheses, participants were sometimes recruited on the basis of these extreme values (i.e., overrepresenting participants with high BMI), thus amplifying the artefact. Without using random sampling techniques and including control groups, studies cannot account for this effect.

In contrast, the qualitative synthesis found that most participants experienced an increase in appetite following the initiation of antipsychotics. However, most participants were purposively sampled for their experience of increased appetite or weight gain, thus limiting the possibility of investigating contradictory experiences. Although the qualitative evidence lacked conceptual richness, it provided a glimpse into the complex interaction between individual, interpersonal and external factors, the most significant being the food environment, in determining whether increased appetite was translated into increased food consumption.

### 6.2. Between-group comparisons

**6.2.1. Summary of evidence.** Overall, there was very low-level observational study evidence that antipsychotic-treated patients with high baseline BMI had higher hunger scores (second-generation antipsychotics) and general food cravings (olanzapine, first-generation antipsychotics) than unexposed healthy participants. Regardless of baseline BMI, there was very low-level observational study evidence that antipsychotics had an inconsistent effect on daily energy intake, increased the likelihood of developing binge eating disorder (second-generation antipsychotics), and had a positive effect on restraint scores, compared to unexposed groups. Antipsychotics had an inconsistent effect on disinhibition and hunger scores in patients with high baseline BMI, while having a positive effect in those with normal baseline BMI.

Using GRADE there was low to very low-level RCT evidence in healthy volunteers with normal baseline BMI that olanzapine or risperidone-treated participants had higher appetite and hunger scores than controls. Olanzapine-treated participants had higher daily energy intake, while aripiprazole-treated participants had lower energy intake, compared to placebo.

**6.2.2. Interpretation and limitations of the evidence.** The aim of a comparative study is to determine the effect of a treatment by comparing outcomes of interest in comparable groups of individuals. These comparison groups should be representative of the target population and comparable in every respect except for the exposure of interest. These studies provide direct meaningful evidence, the most effective of which are placebo-controlled trials with masked allocation. Even though well-conducted placebo-controlled trials ensure assay sensitivity, are scientifically sound and interpretable in terms of efficacy, using placebos has been criticised in clinical situations where effective therapies are available, such as in schizophrenia [117]. This may explain why the comparison groups in most RCTs in this review took active (alternative) agents. However, these head-to-head comparisons answered a different question to the one we were interested in. They compared the effect of different antipsychotics on eating-related outcomes, while we were interested in determining whether antipsychotics had an effect on eating-related outcomes or not.

Consequently, we could not identify direct evidence from "high-quality" studies, in terms of our aims. The majority of between-group comparisons in this review relied on observational studies that examined non-comparable groups. These studies compared eating-related outcomes of antipsychotic-treated participants and unexposed healthy participants from nationally representative [53,56,77] or community samples [27,28,54,58,59,63,75,81,91,101]. Only one observational study [29] compared eating cognitions of antipsychotic-treated and untreated schizophrenia patients. Thus, it is imperative to acknowledge that evidence based on the between-group comparisons in this review is misleading and must be interpreted with great caution. There are well-documented differences between schizophrenia patients and the general population in terms of their higher risk of metabolic abnormalities. People diagnosed with schizophrenia have a two-fold increased risk of metabolic syndrome compared with the general population [20]. This higher risk has been replicated among antipsychotic-naïve patients, hence denoting the possibility of a genetic predisposition to metabolic abnormalities among people diagnosed with schizophrenia regardless of antipsychotic treatment [14,20].

## 7. Limitations and strengths of the review

The internal validity of a review is partly dependant on the quality of the included studies and the credibility of their findings. The quality of included studies in this review was compromised by shortcomings in sample selection and outcome measurement, resulting in confounding, selection bias and information bias that distorted the results.

The included randomised trials were undermined by deficiencies in reporting, especially of their randomisation, concealment of allocation techniques, blinding, and by attrition bias. The non-randomised studies were undermined by their high risk of selection bias, using non-comparable comparison groups, and failure to account for potentially significant confounders. Overall, the studies used small sample sizes and failed to provide the rationale behind sample size calculations.

Quantitative studies included in this review were at high risk of different forms of information bias, including measurement error, recall and social desirability bias. Appetite was mainly assessed using visual analogue scales [27,70,71,78,93,94,99]. Despite evidence suggesting the reliability and reproducibility of visual analogue scales in controlled settings [118], these scales have not been psychometrically tested in the less stringent settings used in most studies included in this review. Furthermore, subjective appetite was primarily measured as a composite variable, but the underlying constructs varied across the studies, thus hampering the interpretation of the evidence. Similarly, eating behaviour traits were measured using different questionnaires, the most common of which was the Three-Factor Eating Questionnaire [119]. This scale has been criticised for its construct validity, factor structure, and factor stability [120–123]. Furthermore, the multitude of fairly similar constructs measured by the different eating behaviour trait questionnaires has contributed in the segregation of the extracted evidence.

The majority of studies measured food intake in natural settings using self-report dietary intake measures. These included short-term measures, such as dietary recall [51,56,67] or food records [76,77], and long-term measures, such as food-frequency questionnaires [48,51–53,56,60,75,77,89–91,98]. These measures have different limitations. Recall measures rely on memory, hence are subject to recall bias [124]. Food records are subject to social desirability bias and reactivity that may lead to subsequent changes in dietary intake [125]. Food frequency questionnaires have a limited list of foods and beverages that can be queried, and are prone to the inaccuracies of the nutrient-composition databases used to convert the collected data into actual food intake [124]. Moreover, under-reporting of energy intake is especially pertinent among people with high BMI [126], thus increasing the risk of misclassification bias.

Errors in outcome measurement are especially problematic when misclassification is systematically different in the groups to be compared. It is possible to assume that in between-group comparisons, antipsychotic-treated participants may have reported outcomes with a different degree of accuracy to those in unexposed groups due to prior knowledge of the association between antipsychotics and weight gain. If this occurred, it may have led to differential misclassification. Additionally, in some studies ascertainment or measurement of the same outcome differed between the exposed (antipsychotic-treated) and unexposed groups [53,56,77], thus compounding misclassification bias.

An additional concern is the measurement of eating-related outcomes as proxies for actual food intake. Visual analogue scales for appetite measurement [118] have not been found to predict energy intake [127]. Self-report dietary data have been heavily criticised for their consistent under-reporting of energy intake, a fact that has been substantiated by evidence from comparisons of self-reported energy intakes to total energy expenditure using doubly-labelled water [124].

The qualitative evidence could not be used to corroborate the quantitative evidence, mainly due to its primary focus on experiences of antipsychotic-induced weight gain and its management. This may also explain why the qualitative evidence lacked conceptual richness. Furthermore, only one qualitative study was conducted in a non-Western setting with limited food availability. That distinctive contextual setting highlighted the significant impact of food environment on patients' experiences of increased appetite. The lack of evidence

from non-Western settings was also replicated in the quantitative evidence, thus revealing a noticeable gap in literature from these settings and limiting the generalisability of the review findings.

The validity of a review is also dependant on the methods used to conduct the review itself. All stages of this review were conducted independently by more than one author to ensure the rigour and credibility of the findings. The GRADE and GRADE-CERQual frameworks were used to enhance the transparency of our judgement of the certainty of the evidence. Incorporating lived experience perspectives into different stages of this review ensured the review questions were relevant to the target population and provided nuance to the interpretation of the review findings. For quantitative synthesis, aggregating effect estimates was not possible owing to the limited number of studies investigating each eating-related outcome; differences between these studies in terms of study design, the operationalisation of eating-related outcomes of interest (in terms of constructs included, measurement and follow-up period), sample characteristics (first episode of drug-naive patients or chronic patients), study setting (inpatient, community/outpatient), antipsychotic agent under study, dose and duration of treatment and previous antipsychotic treatment; and high risk of bias in the results. Vote counting based on direction of effect allowed us to synthesise the quantitative evidence when a meta-analysis of effect estimates was not possible, and gave us a bird's-eye view of the existing literature. Nevertheless, this method neither provides information on the magnitude of effect nor accounts for differences in the relative sizes of the studies [38]. It also relies on creating a standardised binary metric. Using clinically significant differences in eating-related outcomes to define positive and negative effects may have enhanced the interpretability of the quantitative evidence. However, literature in this field has heavily relied on statistical rather than clinical significance when interpreting data. Thus, we had to define positive and negative effects based on values greater or less than zero for mean difference estimates, or greater or less than one for ratio effect estimates, which is problematic as these differences may not necessarily be clinically significant. All together, these factors have limited the interpretation of the review findings.

## 8. Conclusions

We assert that rigorous randomised placebo-controlled trials that are adequately powered to detect clinically significant associations are necessary for establishing the true effects of antipsychotics on eating-related outcomes. Future observational studies, despite providing lower certainty evidence, should include internal comparison groups of antipsychotic-naïve patients, clearly outline the selected sample frame, use probabilistic sampling strategies and collect data on and account for potential confounding and moderating variables in data analysis. To minimise measurement error, future studies should consider the use of the least biased and optimal combination of lab-based, objective and self-report, subjective measures of eating-related outcomes. Similarly, future qualitative research should focus on deepening and enriching our understanding of people's experiences with antipsychotics in terms of their impact on eating and people's relationship with food.

In conclusion, this review has highlighted the paucity of high-quality quantitative evidence on the effect of antipsychotics on eating-related outcomes and qualitative evidence primarily focusing on people's experiences of these effects. This is surprising given the distress reported by service users as a result of these experiences and their possible consequences on mental and physical health. One explanation for this significant gap in research is the lack of involvement of those with lived experience in the existing research. A better understanding of these effects is key to demystifying the mechanisms underlying antipsychotic-induced weight gain, and to developing interventions that support people receiving antipsychotics in managing these effects.

## 9. Lived experience advisory panel commentary

Written by Asmal, S. and C. Hamilton

We all agree that the existing data that was available for our review was of very poor quality. There have been so few qualitative studies and nothing focused primarily on this issue enough and as such goes to show that there is a massive gap in research which still needs a lot of work. This is a topic that hasn't been given the sensitivity, care and attention it so desperately needs. We are so pleased that SUSTAIN is looking into it at this level despite not having better data. In our experience, the intense feeling of intolerable hunger and not getting a sense of feeling full no matter how much food was being consumed were the two most difficult side effects. Following that being told the medication must continue and sometimes in contrast with the stark availability of food and having to continue with it regardless was very hard. Personally, the weight gain and no prior knowledge of how my relationship with food might be affected meant that I stopped accepting the medication which resulted in hospital readmission. We have found that not being told anything about how the medication we are being prescribed may affect our relationship with food as a side effect meant that the weight gain and feelings of intolerable hunger came as a shock. Our recommendation would at a very minimum be that there is an urgent need of a separate 'leaflet' focusing only on the impact of the medication on 'hunger'/ 'relationship with food'. This we believe should be written by the SUSTAIN LEAP members and would explain that once started on antipsychotic medication it could affect the patients' relationship with food and it would highlight that it could and often does result in weight gain. We would include our lived experiences, and this could help carers and patients remain more in control, and more open and honest conversations could be had with health professionals.

## Supporting information

**S1 File. PRISMA checklist.**
(DOCX)

**S2 File. Information sources and search strategy for the systematic review.**
(DOCX)

**S3 File. Data extraction forms.**
(XLSX)

**S4 File. Methods used to prepare the quantitative data for synthesis.**
(DOCX)

**S5 File. Study and participant characteristics of the 55 included quantitative studies.**
(DOCX)

**S6 File. Studies retrieved for full-text examination, including those that were excluded from the analyses and reasons for exclusion.**
(DOCX)

**S7 File. Scales/questionnaires used to assess eating-related outcomes.**
(DOCX)

**S8 File. Risk of bias assessments of the 55 included quantitative studies using the Mixed Methods Assessment Tool.**
(DOCX)

**S9 File. Detailed account of the quantitative syntheses.**
(DOCX)

**S10 File. The GRADE Evidence Profile for key outcomes.**
(DOCX)

**S11 File. Study and participant characteristics of the 6 included qualitative studies.**
(DOCX)

**S12 File. Risk of bias assessments of the 6 included qualitative studies using the Mixed Methods Appraisal Tool (MMAT).**
(DOCX)

**S13 File. The GRADE-CERQual assessment of confidence in qualitative evidence.**
(DOCX)

## Acknowledgments

The authors would like to thank Dr Sarah Rhodes (Senior Lecturer in Biostatistics, at the Institute of Population Health, The University of Manchester, UK) for her advice on quantitative synthesis approaches; and Dr Gill Norman (Principal Research Associate, Newcastle University NIHR Innovation Observatory, UK) and Dr Chunhu Shi (Research Fellow, at the School of Health Sciences, The University of Manchester, UK) for their insightful comments on an initial draft of this work. We would also like to thank Ms Claire Hodkinson, The University of Manchester specialist librarian, for advising on the design of search strategies.

## Author contributions

**Conceptualization:** Karina Lovell, Penny Bee, Rebecca Pedley, Helen Louise Brooks, Richard J Drake, Prathiba Chitsabesan, Andrew Grundy.

**Data curation:** Rasha Alkholy.

**Formal analysis:** Rasha Alkholy.

**Funding acquisition:** Karina Lovell, Penny Bee.

**Investigation:** Rasha Alkholy, Karina Lovell, Penny Bee, Rebecca Pedley, Helen Louise Brooks, Anam Bhutta, Abigail Brown, Rebecca L Jenkins, Andrew Grundy.

**Methodology:** Karina Lovell, Penny Bee, Rebecca Pedley, Helen Louise Brooks, Richard J Drake.

**Supervision:** Karina Lovell, Penny Bee, Rebecca Pedley, Richard J Drake, Andrew Grundy.

**Visualization:** Rasha Alkholy.

**Writing – original draft:** Rasha Alkholy.

**Writing – review & editing:** Rasha Alkholy, Karina Lovell, Penny Bee, Rebecca Pedley, Helen Louise Brooks, Richard J Drake, Prathiba Chitsabesan, Anam Bhutta, Andrew Grundy.

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
