## [Decision Letter · Decision Letter 0]

19 Nov 2024

PONE-D-24-29459The impacts of antipsychotic medications on eating experiences and eating behaviours: A mixed methods systematic reviewPLOS ONE

Dear Dr. Alkholy,

Thank you for submitting your manuscript to PLOS ONE. After careful consideration, we feel that it has merit but does not fully meet PLOS ONE’s publication criteria as it currently stands. Therefore, we invite you to submit a revised version of the manuscript that addresses the points raised during the review process.

**ACADEMIC EDITOR: ** I am giving you an opportunity to address the minor typographical errors identified by the reviewer and thoroughly proofread the manuscript before I make a final decision on your manuscript.

If applicable, we recommend that you deposit your laboratory protocols in protocols.io to enhance the reproducibility of your results. Protocols.io assigns your protocol its own identifier (DOI) so that it can be cited independently in the future. For instructions see: https://journals.plos.org/plosone/s/submission-guidelines#loc-laboratory-protocols . Additionally, PLOS ONE offers an option for publishing peer-reviewed Lab Protocol articles, which describe protocols hosted on protocols.io. Read more information on sharing protocols at https://plos.org/protocols?utm_medium=editorial-email&utm_source=authorletters&utm_campaign=protocols.

We look forward to receiving your revised manuscript.

Kind regards,

Anthony A. Olashore, MBCHB, PhD, FWACP

Academic Editor

PLOS ONE

Journal Requirements:

2. Thank you for stating the following financial disclosure: [This paper presents research funded by the National Institute for Health and Care Research (NIHR) Applied Research Collaboration Greater Manchester (ARC-GM); Grant number NIHR200174 (https://arc-gm.nihr.ac.uk/mental-health). KL is a co-applicant on the ARC award and the Mental Health Theme Lead. PB is the Deputy Theme Lead. The views expressed are those of the authors and not necessarily those of the NIHR or the Department of Health and Social Care.]. Please state what role the funders took in the study. If the funders had no role, please state: "The funders had no role in study design, data collection and analysis, decision to publish, or preparation of the manuscript." If this statement is not correct you must amend it as needed. Please include this amended Role of Funder statement in your cover letter; we will change the online submission form on your behalf.

3. Please include captions for your Supporting Information files at the end of your manuscript, and update any in-text citations to match accordingly. Please see our Supporting Information guidelines for more information: http://journals.plos.org/plosone/s/supporting-information .

4. As required by our policy on Data Availability, please ensure your manuscript or supplementary information includes the following:

Reviewers' comments:

Reviewer's Responses to Questions

**Comments to the Author**

1. Is the manuscript technically sound, and do the data support the conclusions?

Reviewer #1: Yes

Reviewer #2: Yes

2. Has the statistical analysis been performed appropriately and rigorously? 

Reviewer #1: N/A

Reviewer #2: Yes

3. Have the authors made all data underlying the findings in their manuscript fully available?

Reviewer #1: Yes

Reviewer #2: Yes

4. Is the manuscript presented in an intelligible fashion and written in standard English?

Reviewer #1: Yes

Reviewer #2: Yes

5. Review Comments to the Author

Reviewer #1: The systematic review manuscript is scientifically sound with conclusions that are well supported by the data findings from the reviewed studies.

There was no statistical analysis required in the systematic review. Authors utilised vote counting on the direction of effects in view of the quality of studies regarding power.

Manuscript was presented in standard English. There are only two typographic errors noted by this reviewer in the write-up: Line 176 (page 8) “trails” for ‘trials’, and line 285 (page 13) “13,13,502” for “13,502”.

In sum, the manuscript presented a procedurally detailed mixed methods systematic review of the impacts of antipsychotic medications on eating experiences and eating behaviour of subjects, with adherence to up-to-date best practice guidelines.

The findings are germane, with appropriate interpretation, discussions and inferences on the existing quantitative and qualitative studies, on the subject. The involvement of a lived experience advisory panel adds to the validity of the review, with good recommendations for future studies.

Reviewer #2: This is a well researched and technically sound paper that highlights the paucity of high level evidence on an important topic. Reading the title one would think that the paper will highlight some distinction between outcomes related to eating experiences and eating behaviours hence the need to explicitly specify them in the title, however there is no distinction in the body of the paper between the two. I would therefore suggest that perhaps a better title would be "The impacts of antipsychotic medications on eating related outcomes: A mixed methods systematic review"

6. PLOS authors have the option to publish the peer review history of their article (what does this mean? ). If published, this will include your full peer review and any attached files.

**Do you want your identity to be public for this peer review?** For information about this choice, including consent withdrawal, please see our Privacy Policy .

Reviewer #1: No

Reviewer #2: No

---

## [Author Response · Author response to Decision Letter 1]

31 Dec 2024

Manuscript Number: PONE-D-24-29459 

The impacts of antipsychotic medications on eating experiences and eating behaviours: A mixed methods systematic review

Response to Reviewers

Dear Dr Olashore,

Thank you for giving us the opportunity to submit a revised version of the manuscript “The impacts of antipsychotic medications on eating experiences and eating behaviours: A mixed methods systematic review” for publication in PLOS ONE.

We appreciate the time and effort that you and the reviewers dedicated in providing constructive feedback on our manuscript and are grateful for the insightful comments and valuable improvements to our paper.

We have incorporated the suggested revisions and amended the manuscript accordingly. The amendments are highlighted (in yellow) within the revised manuscript file with tracked changes (file name: Revised Manuscript with Track Changes). An unmarked version of the revised manuscript, without tracked changes, has also been submitted (file name: Manuscript).

Please see below for a response to each point raised by the Academic Editor and Reviewers. Amended text is highlighted in yellow in the 'Revised Manuscript with Track Changes' file. All page, paragraph and line numbers refer to the revised manuscript file with tracked changes (file name: Revised Manuscript with Track Changes).

Academic Editor Comments:

https://journals.plos.org/plosone/s/file?id=wjVg/PLOSOne_formatting_sample_main_body.pdf [journals.plos.org] and

https://journals.plos.org/plosone/s/file?id=ba62/PLOSOne_formatting_sample_title_authors_affiliations.pdf [track.editorialmanager.com]

Author response:

Thank you very much for reviewing our manuscript and providing your constructive feedback. We deeply appreciate your time, effort and valuable guidance.

We have amended the manuscript and file naming to meet PLOS ONE’s style requirements, as requested:

•Figure in-text citations, captions and files:

We have amended the in-text figure citations (as Fig 1, Fig 2 etc) and the corresponding figure captions (labels, titles, legends) as per PLOS ONE’s guidelines.

The numbering order of the figures has been revised, and all corresponding figure citations, captions, and file names have been updated accordingly.

We have used the PACE tool to generate the figure files in TIFF format and to ensure adherence to PLOS ONE’s file requirements.

Figure files have been uploaded separately as individual files.

The Response to Reviewers file includes a detailed table (including page and line numbers) of amended figure in-text citations, captions and filing names.

•Author affiliations:

We have amended the author affiliations as per PLOS ONE’s ‘Title, author and affiliations formatting guidelines’.

The Response to Reviewers file includes a detailed table (including page and line numbers) of amended author affiliations.

•We have adhered to PLOS ONE’s heading and manuscript body formatting guidelines.

Academic Editor Comments to the Authors:

2. Thank you for stating the following financial disclosure: [This paper presents research funded by the National Institute for Health and Care Research (NIHR) Applied Research Collaboration Greater Manchester (ARC-GM); Grant number NIHR200174 (https://arc-gm.nihr.ac.uk/mental-health [arc-gm.nihr.ac.uk]). KL is a co-applicant on the ARC award and the Mental Health Theme Lead. PB is the Deputy Theme Lead. The views expressed are those of the authors and not necessarily those of the NIHR or the Department of Health and Social Care.]. Please state what role the funders took in the study. If the funders had no role, please state: "The funders had no role in study design, data collection and analysis, decision to publish, or preparation of the manuscript." If this statement is not correct you must amend it as needed. Please include this amended Role of Funder statement in your cover letter; we will change the online submission form on your behalf.

Author response:

Thank you very much for your constructive feedback.

We confirm that the funders had no role in this study. We have added the Role of the Funder statement provided by the Academic Editor in the Cover Letter, as requested: "The funders had no role in study design, data collection and analysis, decision to publish, or preparation of the manuscript."

Academic Editor Comments to the Authors:

3. Please include captions for your Supporting Information files at the end of your manuscript, and update any in-text citations to match accordingly. Please see our Supporting Information guidelines for more information: http://journals.plos.org/plosone/s/supporting-information [track.editorialmanager.com].

Author response:

Thank you very much for your constructive feedback.

We have amended the Supporting Information files in-text citations, captions and file names to meet PLOS ONE’s Supporting Information guidelines:

•We have included captions for all Supporting Information files at the end of the manuscript (after the References section; including the file name, number and title).

•We have updated in-text citations of all Supporting Information files accordingly.

•Supporting information files have been uploaded separately as individual files.

•The Response to Reviewers file includes a detailed table (including page and line numbers) of amended Supporting information files, in-text citations, captions and filing names.

Academic Editor Comments to the Authors:

4. As required by our policy on Data Availability, please ensure your manuscript or supplementary information includes the following:

Author response:

Thank you very much for your constructive feedback. The location of the requested information within the main text and the supplementary information files is indicated in the following section.

• Academic Editor Comments:

Author response:

The requested table is provided in the S6 File. This file incudes a numbered table of all studies retrieved for full-text examination, including those that were excluded from the analyses and reasons for exclusion.

• Academic Editor Comments:

Author response:

The requested information is provided in the S6 File.

• Academic Editor Comments:

Author response:

All of the studies included in the systematic review are published.

• Academic Editor Comments:

o Name of data extractors and date of data extraction

o Confirmation that the study was eligible to be included in the review.

o All data extracted from each study for the reported systematic review and/or meta-analysis that would be needed to replicate your analyses.

o If data or supporting information were obtained from another source (e.g. correspondence with the author of the original research article), please provide the source of data and dates on which the data/information were obtained by your research group.

Author response:

The requested table is provided in the S3 File. This file includes tables of all data extracted from the records included in the systematic review including: name of data extractors, date of data extraction, confirmation that the study was eligible to be included in the review, all data extracted from each study for the reported systematic review that would be needed to replicate the analyses. When data or supporting information were obtained from other relevant records, this was clearly indicated and references for these records (and dates of access) were provided.

• Academic Editor Comments:

Author response:

The S8 File includes the completed risk of bias assessments of the 55 included quantitative studies using the Mixed Methods Assessment Tool (MMAT). The S12 File includes the completed risk of bias assessments of the 6 included qualitative studies using the Mixed Methods Appraisal Tool (MMAT). The GRADE Summary of Findings Table for key outcomes for within-group syntheses is presented in Fig 6. The GRADE Summary of Findings Table for key outcomes for between-group syntheses is presented in Fig 11. The S10 File incudes the GRADE Evidence Profile for each key outcome. It contains information about the body of evidence, judgments about each of the quality of evidence factors, key results (using vote counting with direction of effect), and the quality of evidence rating for each outcome. The GRADE-CERQual Summary of Qualitative Findings is presented in Table 1. The S13 File incudes the GRADE-CERQual Evidence Profile for each qualitative synthesis finding.

• Academic Editor Comments:

Author response:

A detailed explanation of the methods used to prepare the data for synthesis, including data conversion and handling of missing summary statistics is provided in the S4 File. Overall, data required for the within-group and between-group syntheses were provided in numerical format. Occasionally, required data were presented in graphical format. In these cases, lead authors of the respective records were contacted via email and requested to provide the missing data. In the absence of a response, the graphical representation of results was used to determine the direction of effect. When this approach was employed, it was clearly indicated in the detailed ‘Summary of effect measures of antipsychotic medications on eating-related outcomes’ tables and ‘GRADE’ tables provided in the Supplementary Files.

Academic Editor Comments

Author response:

Thank you for your constructive feedback. We have not cited any retracted papers in our manuscript. 12 references have been amended to adhere with PLOS ONE’s referencing style (“Vancouver” style). The amended references and reasons for amnedment (eg to adhere to PLOS ONE referencing style requirements, adding a missing DOI) are provided in a table in the Response to Reviewers file.

Reviewer 1 Comments:

The systematic review manuscript is scientifically sound with conclusions that are well supported by the data findings from the reviewed studies.

There was no statistical analysis required in the systematic review. Authors utilised vote counting on the direction of effects in view of the quality of studies regarding power.

Manuscript was presented in standard English. There are only two typographic errors noted by this reviewer in the write-up: Line 176 (page 8) “trails” for ‘trials’, and line 285 (page 13) “13,13,502” for “13,502”.

In sum, the manuscript presented a procedurally detailed mixed methods systematic review of the impacts of antipsychotic medications on eating experiences and eating behaviour of subjects, with adherence to up-to-date best practice guidelines.

The findings are germane, with appropriate interpretation, discussions and inferences on the existing quantitative and qualitative studies, on the subject. The involvement of a lived experience advisory panel adds to the validity of the review, with good recommendations for future studies.

Author response:

Thank you very much for reviewing our paper and providing your constructive feedback. We are extremely grateful for your time, effort and valuable guidance.

We apologise for the typographical errors and express our sincere gratitude to the Reviewer for identifying them.

We have amended the manuscript accordingly:

Page 3, line number 46: 'trails' has been amended to 'trials'.

Page 10, line number 177: 'trails' has been amended to 'trials'.

Page 14, line number 289: “13,13,502” has been amended to “13,502”.

(N.B.: The original line numbers and pages have been revised following the incorporation of all amendments.)

Reviewer 2 Comments:

This is a well researched and technically sound paper that highlights the paucity of high level evidence on an important topic. Reading the title one would think that the paper will highlight some distinction between outcomes related to eating experiences and eating behaviours hence the need to explicitly specify them in the title, however there is no distinction in the body of the paper between the two. I would therefore suggest that perhaps a better title would be "The impacts of antipsychotic medications on eating related outcomes: A mixed methods systematic review".

Author response:

Thank you very much for reviewing our paper and providing your constructive feedback. We are extremely grateful for your time, effort and valuable guidance.

We agree with the Reviewer’s recommendations regarding the title. Using ‘eating-related outcomes’ instead of ‘eating experiences and eating behaviours’ avoids potential confusion and more comprehensively encompasses the wide spectrum of eating-related outcomes. We have amended the full title and short title of this manuscript accordingly.

The following sections in the Title page have been amended accordingly.

Full title: The impacts of antipsychotic medications on eating-related outcomes: A mixed methods systematic review

Short title: Antipsychotic medications and eating-related outcomes: A mixed methods systematic review

We wish to express our sincere gratitude to the Academic Editor and the Reviewers for their valuable guidance. We look forward to receiving your response regarding our revised submission and will gladly address any questions or comments you may have.

Thank you.

---

## [Editor Report · Decision Letter 1]

8 Jan 2025

The impacts of antipsychotic medications on eating-related outcomes: A mixed methods systematic review

PONE-D-24-29459R1

Dear Dr. Alkholy,

We’re pleased to inform you that your manuscript has been judged scientifically suitable for publication and will be formally accepted for publication once it meets all outstanding technical requirements.

Kind regards,

Anthony A. Olashore, MBCHB, PhD, FWACP

Academic Editor

PLOS ONE
---

## [Editor Report · Acceptance letter]

PONE-D-24-29459R1

PLOS ONE

Dear Dr. Alkholy,

I'm pleased to inform you that your manuscript has been deemed suitable for publication in PLOS ONE. Congratulations! Your manuscript is now being handed over to our production team.

Kind regards,

on behalf of

Dr. Anthony A. Olashore

Academic Editor

PLOS ONE